



# On the importance of atmospheric loss of organic nitrates by aqueous-phase ·OH-oxidation

Juan Miguel González-Sánchez[1,2], Nicolas Brun[1,2], Junteng Wu[1], Julien Morin[1], Brice Temime-Rousell[1], Sylvain Ravier[1], Camille Mouchel-Vallon[3], Jean-Louis Clément[2], Anne Monod[1]

[1] Aix Marseille Univ, CNRS, LCE, Marseille, France
    [2] Aix Marseille Univ, CNRS, ICR, Marseille, France
    [3] Laboratoire d'Aérologie, Université de Toulouse, CNRS, UPS, Toulouse, France

*Correspondence to*: Juan Miguel González-Sánchez (juanmiguelgs93@gmail.com) and Anne Monod (anne.monod@univ-amu.fr)

**Abstract.** Organic nitrates are secondary species in the atmosphere. Their fate is related to the chemical transport of pollutants from polluted areas to more distant zones. While their gas-phase chemistry has been studied, their reactivity in condensed phases is far from being understood. However, these compounds represent an important fraction of organic matter in condensed phases. In particular, their partition to the aqueous-phase may be especially important for oxidized organic nitrates for which water solubility increase with functionalization. This work has studied for the first time the aqueous-phase ·OH-oxidation

kinetics of 5 alkyl nitrates (isopropyl nitrate, isobutyl nitrate, 1-pentyl nitrate, isopentyl nitrate and 2-ethylhexyl nitrate) and 3 functionalized organic nitrates (α-nitrooxyacetone, 1-nitrooxy-2-propanol and isosorbide 5-mononitrate) by developing a novel and accurate competition kinetic method. Low reactivity was confirmed, with $k_{OH}$ (at $296 \pm 2$ K) ranging from $8 \cdot 10^7$ to $2.5 \cdot 10^9$ L mol$^{-1}$ s$^{-1}$. Using these results, the previously developed aqueous-phase Structure Activity Relationship (SAR) was extended, and the resulting parameters confirmed the extreme deactivating effect of the nitrate group, up to two adjacent carbon

atoms. The achieved extended SAR was then used to determine the ·OH-oxidation rate constants of 49 organic nitrates, including hydroxy nitrates, ketonitrates, aldehyde nitrates, nitrooxy carboxylic acids and more functionalized organic nitrates such as isoprene and terpene nitrates. Their multiphase atmospheric lifetimes towards ·OH-oxidation were calculated using these rate constants, and compared to their gas-phase lifetimes. Large differences were observed, especially for polyfunctional organic nitrates: for 50% of the proposed organic nitrates for which ·OH-reaction occurs mainly in the aqueous-phase (more than 50% of the overall removal) their ·OH-oxidation lifetimes increased by 20% to 155% under cloud/fog conditions ($L_{WC} =$

0.35 g m$^{-3}$). In particular, for 83% of the proposed terpene nitrates, the reactivity towards ·OH occurred mostly ($> 98\%$) in the aqueous-phase while for 60% of these terpene nitrates their lifetimes increased by 25% to 140% compared to their gas-phase reactivity. We demonstrate that these effects are of importance under cloud/fog conditions, but also under wet aerosol conditions, especially for the terpene nitrates. These results suggest that taking into account aqueous-phase ·OH-oxidation

reactivity of biogenic nitrates is necessary to improve the predictions of their atmospheric fate.



# 1 Introduction

Nitrogen oxides ($NO_x = \cdot NO + \cdot NO_2$) intensely impact air quality and the environment as they play key roles in the production of relevant air pollutants such as ozone ($O_3$), nitrous acid (HONO), nitric acid ($HNO_3$) and secondary organic aerosol (SOA). Their atmospheric chemistry controls the concentrations of the three main oxidants, $O_3$, $\cdot OH$ and $NO_3\cdot$ radicals. The past few decades have witnessed important reduction in $NO_x$ direct emissions in Europe or North America resulting in changes in their atmospheric fate, by increasing the relative importance of their conversion to organic nitrates (Romer Present et al., 2020). The latter are secondary organic compounds, formed by the reactivity of $NO_x$ with volatile organic compounds (VOCs). Owing to their long atmospheric lifetimes (much longer than for $NO_x$), organic nitrates can be transported from polluted to remote areas. During their long-range transport, these compounds undergo photolysis and/or $\cdot OH$-oxidation resulting in the release of $NO_x$ far from the $NO_x$ sources. Organic nitrates thus act as sinks and reservoirs of $NO_x$, leading to a broader spatial distribution of $NO_x$ and thus spreading the ozone, HONO and SOA formation from the local to the regional scale. For this reason, understanding and considering the reactivity of organic nitrates is necessary for accurately predicting their atmospheric fates and impacts on air quality.

The gas-phase chemistry of organic nitrates has been studied through kinetic experiments focusing on their $\cdot OH$-oxidation (Bedjanian et al., 2018; Picquet-Varrault et al., 2020; Treves and Rudich, 2003; Wängberg et al., 1996; Zhu et al., 1991) and direct photolysis (Clemitshaw et al., 1997; Picquet-Varrault et al., 2020; Suarez-Bertoa et al., 2012). These experiments provide data for different types of organic nitrates, including alkyl nitrates, ketonitrates, hydroxy nitrates, dinitrates, cyclonitrates and alkene nitrates, and provide knowledge on their gas-phase atmospheric fate. Although alkyl and alkene nitrates are highly volatile, polyfunctional organic nitrates may show much lower volatility and they can partition to condensed phases (aqueous- and aerosol-phase). Their presence in submicron particles has been observed in a fraction ranging from 5 to 77 % (in mass) of organic aerosol in Europe and North America (Kiendler-Scharr et al., 2016; Lee et al., 2019). Despite this fact, their reactivities in condensed phases were poorly explored. Most studies have focused on hydrolysis, a reaction that is extremely structure dependent, mostly occurring to tertiary nitrates (Hu et al., 2011; Liu et al., 2012), while the non-hydrolyzable fraction of α- and β-pinene particulate organic nitrate range from 68 to 91 % in mass (Takeuchi and Ng, 2019). To our knowledge, only one study experimentally determined the photolysis kinetics of organic nitrates in the aqueous-phase (Romonosky et al., 2015) and concluded that the $\cdot OH$ removal processes should have a higher relevance. However, there has been no attempts to experimentally determine their aqueous-phase $\cdot OH$-oxidation reactivity.

Aqueous-phase $\cdot OH$-oxidation processes play a key role in the removal and the production of water soluble compounds in the atmosphere (Herrmann et al., 2015). The determination of kinetic rate constants is essential to understand their lifetimes and to develop more precise models to predict pollution events and the scale of pollutants' transportation. Determining aqueous-phase $\cdot OH$-oxidation second order rate constants ($k_{OH}$) may be done by a direct method, measuring $\cdot OH$ concentrations in the aqueous-phase by UV-VIS absorption, with however troublesome interferences of absorbance of organic molecules within the same wavelength region (Herrmann et al., 2010). Indirect methods have been then widely developed, consisting in the





comparison between the decay of the target compound versus a reference compound, for which $k_{OH}$ value is well-known. Many studies have used the thiocyanate anion (SCN⁻) as the reference compound. Its decay can be easily followed by online spectroscopic measurement of the formed radical anion $(SCN)_2^{\cdot-}$ in presence and in absence of the target compound (Herrmann, 2003). This technique, however, is not well suited for low reactive species for which $k_{OH} \leq 10^9$ L mol⁻¹ s⁻¹, (i.e. more than 10 times lower than $k_{OH + SCN-} = 1.12 \, (\pm 0.20) \cdot 10^{10}$ L mol⁻¹ s⁻¹). In these cases, the target species should reach high concentrations to be able to compete with the thiocyanate anion. These high concentrations may interfere with the measurements. Furthermore, the uncertainty on $k_{OH + SCN-}$ may be higher than the target compound rate constant. Other indirect methods use offline measurements for monitoring the target and reference compounds. However, most of these methods use reference compounds for which $k_{OH}$ values have not been so widely explored thus inducing important uncertainties to the experimentally determined $k_{OH}$ values (Herrmann, 2003).

The aim of this work was to accurately determine aqueous-phase $k_{OH}$ rate constants by developing a new online competition kinetic method, well suited to low reactivity species. The effectiveness and the relevance of our method was validated on compounds for which $k_{OH}$ rate constants are well-known. Then, the method was used to determine the $k_{OH}$ rate constants of some organic nitrates and the results were used to extend the aqueous-phase Structure Activity Relationship (SAR) developed earlier (Monod and Doussin, 2008 and Doussin and Monod, 2013) to the nitrate group. Furthermore, the prediction of $k_{OH}$ rate constants for other atmospherically relevant organic nitrates in the aqueous-phase was performed with the extended SAR. Finally, the potential multiphase fate in the atmosphere of these compounds was estimated.

## 2 Experimental

### 2.1 Principle

The originality of this work relies on a competition kinetic method monitored by online decay measurements of the reference compound in the reactor's headspace using a Proton Transfer Reaction-Mass Spectrometer (PTR-MS). Concerning the target compounds, depending on their properties (solubility, volatility and instrumental response), their kinetic decays were monitored in the reactor's headspace by PTR-MS or off-line by Ultra-High-Performance Liquid Chromatography-Photodiode Array detector (UHPLC-UV). The developed method employed methanol as the reference compound because: i) its $k_{OH}$ rate constant, $9.7 \, (\pm 1.5) \cdot 10^8$ L mol⁻¹ s⁻¹ at 298 K, is widely accepted (Table S1); ii) this value is relevant for competitive kinetics relative to low reactive species, and iii) its extremely sensitive quantification by PTR-MS was expected to not suffer from any interferences with the selected organic nitrates (Aoki et al., 2007; Lindinger et al., 1998).

Due to the known sensitivity of organic nitrates to photolysis (Romonosky et al., 2015), aqueous ·OH radicals were generated in the dark by the Fenton reaction (R1) (Neyens and Baeyens, 2003) by dropwise addition of a solution of $Fe^{2+}$ to an acidic solution of $H_2O_2$ in excess containing methanol and the target compound.

$$Fe^{2+} + H_2O_2 \rightarrow Fe^{3+} + \cdot OH + OH^- \, , \tag{R1}$$



The method was validated using isopropanol and acetone as target compounds, whose ·OH-oxidation rate constants are well-known (Table S1). The method was then used to determine new $k_{OH}$ values for five alkyl nitrates (isopropyl nitrate, isobutyl nitrate, 1-pentyl nitrate, isopentyl nitrate and 2-ethylhexyl nitrate) and three polyfunctional organic nitrates (α-nitrooxyacetone, 1-nitrooxy-2-propanol and isosorbide 5-mononitrate) (Table 1).

## 2.2 Experimental setup and protocol

The aqueous-phase reactor consisted in a one-liter three-neck round-bottom flask closed hermetically with rubber caps (Fig. 1). A Razel syringe pump at 0.33 mL min$^{-1}$ was used with a glass syringe to add dropwise a solution of $Fe^{2+}$ into the reactor's aqueous solution. A second syringe was also connected to the reactor to add the target and the reference compound and to sample the aqueous-phase when necessary. The aqueous reactor's solution was continuously stirred before and during the reaction with a magnetic stirrer. A flow of synthetic air constantly guided a fraction of the headspace gas-phase of the reactor

to the PTR-MS Instrument, with a flow of 0.050 L min$^{-1}$ using a mass flow controller (Brooks SLA Series). A 1:30 dilution was performed downward the reactor's flow using synthetic air at 1.450 L min$^{-1}$. All experiments were performed at room temperature ($296 \pm 2$ K).

In each experiment, the $Fe^{2+}$ solution consisted in $5 - 10$ mL of $FeSO_4 \cdot 7H_2O$ ($0.02 - 0.06$ mol L$^{-1}$), added dropwise to the solution of 400 mL of $H_2O_2$ (in excess, i.e. 0.004 mol L$^{-1}$) acidified by $H_2SO_4$ (0.005 mol L$^{-1}$), to keep pH < 3 during the

reaction. Concentrations and volumes of the $Fe^{2+}$ solution were varied in order to optimize the ·OH attack to the reference and the target compound. All experiments and their initial conditions are compiled on Table S2. The $Fe^{2+}$ solution was used as the limiting reagent to minimize its possible interferences with the organic nitrate reaction products. In addition, it was verified in control experiments that $H_2O_2$, used in excess, did not react with any of the target compounds.

Prior each experiment, the reactor's headspace was extensively purged with pure air for 1 hour while PTR-MS measurements

were set for stabilization purposes. The target compound and methanol were then added directly into the aqueous-phase and the reactor's headspace signal was monitored for 30 minutes with constant stirring of the solution to determine the first order rate constant of evaporation ($k_{vap}$) of each compound under the reaction conditions. The reaction was then started by adding dropwise the $Fe^{2+}$ solution for 15 to 30 minutes (depending on the volume of $Fe^{2+}$ added). During experiments performed with isopropyl nitrate, α-nitrooxyacetone, 1-nitrooxy-2-propanol and isosorbide 5-mononitrate, the aqueous-phase was sampled

every three minutes for off-line measurements by UHPLC-UV. Once the addition of $Fe^{2+}$ was stopped, the system was kept stirring for another 30 minutes while continuously monitoring the reactor's headspace.

## 2.3 Methodology

In such a competition between the target compound (X) and methanol (M) towards ·OH radicals (R2),

$$X/M + \cdot OH \rightarrow Products ,\qquad\qquad\qquad\qquad\qquad\qquad\qquad\qquad\text{(R2)}$$





in the absence of any photolysis reaction, the kinetic rate constant of the target compound, $k_{OH,X}$, is directly obtained from the

slope ($k_{OH,X}/k_{OH,M}$) of the linear plot of $\ln([X]_{0\,(aq)}/[X]_{t\,(aq)})$ versus $\ln([M]_{0\,(aq)}/[M]_{t\,(aq)})$ derived from Eq. (1).

$$ln\frac{[X]_{0\,(aq)}}{[X]_{t\,(aq)}} = \frac{k_{OH,X}}{k_{OH,M}} \cdot ln\frac{[M]_{0\,(aq)}}{[M]_{t\,(aq)}}, \qquad (1)$$

where $\ln([X]_{0\,(aq)}/[X]_{t\,(aq)})$ represents the aqueous-phase concentration relative decay of the target compound, and $\ln([M]_{0\,(aq)}/[M]_{t\,(aq)})$ the aqueous-phase concentration relative decay of methanol.

Using the Fenton reaction (R1), ·OH radicals were generated within the aqueous-phase and could not reach the headspace in sufficient amounts to react significantly with the target compounds in the gas-phase. This was confirmed by the validation experiments (section 4.1) where the values found for $k_{OH,X}/k_{OH,M}$ ratios were those of the aqueous-phase reactions, and not those of the gas-phase which are a factor of 2 to 3 higher.

Headspace analyses were based on the direct proportionality between aqueous- and gas-phase concentrations of the analytes

(Karl et al., 2003). Therefore, in cases where both the target and the reference compounds were followed by PTR-MS, Eq. (1) can be written as Eq. (2), (see Appendix A for detailed explanations),

$$ln\frac{ncps(X)_0}{ncps(X)_t} = \frac{k_{OH,X}}{k_{OH,M}} \cdot ln\frac{ncps(M)_0}{ncps(M)_t}, \qquad (2)$$

where $ncps(X)$ and $ncps(M)$ are the normalized count rate of respectively the target compound and methanol, i.e: $ncps(X) = \sum cps(X^+)/cps(H_3O^+)$ and $ncps(M) = \sum cps(M^+)/cps(H_3O^+)$. The normalization to the $H_3O^+$ count rate removes any

signal variability due to fluctuating performances of the ion source.

Depending on the properties of the investigated organic nitrates (i.e. their water solubility, volatility, and instrumental response), different treatments were applied to determine their kinetic decays. All the very volatile compounds were monitored in the reactor's headspace by PTR-MS: this comprised the target alkyl nitrates (for which Henry's Law constant $K_H < 1$ mol $L^{-1}$ atm$^{-1}$), as well as methanol, acetone and isopropanol ($K_H < 300$ mol $L^{-1}$ atm$^{-1}$). Furthermore, due to the low water solubility

of alkyl nitrates, their initial concentrations were limited to $5 \cdot 10^{-5}$ mol $L^{-1}$. On the other hand, the target polyfunctionalized organic nitrates were much less volatile ($K_H > 1000$ mol $L^{-1}$ atm$^{-1}$) and their detection by PTR-MS was more problematic, their relative decays were monitored in the aqueous-phase by off-line UHPLC-UV. For these compounds, the determination of the aqueous-phase ·OH-oxidation rate constant was performed by plotting the relative decay of the aqueous-phase concentration ($ln\,([X]_{0\,(aq)}/[X]_{t\,(aq)})$) against the relative decay of the reference compound signal in the gas-phase

($ln\,(ncps(M)_0/ncps(M)_t)$ as shown in Eq. (3):

$$ln\frac{[X]_{0\,(aq)}}{[X]_{t\,(aq)}} = \frac{k_{OH,X}}{k_{OH,M}} \cdot ln\frac{ncps(M)_0}{ncps(M)_t}, \qquad (3)$$

In addition, for the most volatile molecules (i.e. alkyl nitrates), further corrections were performed to subtract the contribution of evaporation to the overall reaction decay. Their high volatilities induced a substantial decay of their concentrations that was





systematically measured prior the start of the reaction, inferring a specific first order rate constant of evaporation ($k_{vap}$) of
each compound under the reaction conditions. While for methanol, acetone, or isopropanol the evaporation decay was
negligible, for the alkyl nitrates it accounted for 2 to 20% of the molecule's consumption during the reaction. To consider this
contribution, Eq. (4) was used instead of Eq. (2) for alkyl nitrates:

$$\frac{1}{t} \cdot ln\frac{ncps(X)_0}{ncps(X)_t} = k_{vap} + \frac{k_{OH,X}}{k_{OH,M}} \cdot \frac{1}{t} \cdot ln\frac{ncps(M)_0}{ncps(M)_t}, \tag{4}$$

where $t$ is the reaction time (in seconds). Plotting $1/t \cdot ln(ncps(X)_0/ncps(X)_t)$ versus $1/t \cdot ln(ncps(M)_0/ncps(M)_t)$
resulted in a straight line with a slope of $k_{OH,X}/k_{OH,M}$ and an intercept of $k_{vap}$. The value of $k_{vap}$ was determined during the
30 minutes prior the reaction and its value was fixed in Eq. (4). This treatment was not necessary for methanol, acetone or
isopropanol, due to their lower volatilities. For isopropyl nitrate, which decay was also followed in the aqueous-phase Eq. (5)
was used (in addition to Eq. (4)):

$$\frac{1}{t} \cdot ln\frac{[X]_{0\,(aq)}}{[X]_{t\,(aq)}} = k_{vap} + \frac{k_{OH,X}}{k_{OH,M}} \cdot \frac{1}{t} \cdot ln\frac{ncps(M)_0}{ncps(M)_t}, \tag{5}$$

In summary, the aqueous-phase ·OH-oxidation rate constant of isopropanol and acetone was determined using Eq. (2); while
Eq. (3) was used for determining the rate constants of α-nitrooxyacetone, 1-nitrooxy-2-propanol and isosorbide 5-mononitrate.
For isobutyl nitrate, 1-pentyl nitrate, isopentyl nitrate and 2-ethylhexyl nitrate, compounds which undergo a significant
evaporation decay, Eq. (4) was used. Finally, isopropyl nitrate was chosen for intercomparing both methods as it is water
soluble and volatile enough to be monitored by both analytical techniques, its aqueous-phase $k_{OH}$ was determined by both Eq.
(4) and Eq. (5).

### 2.4 Analytical measurements

#### 2.4.1 PTR-MS

A commercial high-sensitivity quadrupole PTR-MS (Ionicon Analytik GmbH) was used to monitor the concentration decay
of the reference compound (methanol) and 7 target compounds (acetone, isopropanol, isopropyl nitrate, isobutyl nitrate, 1-
pentyl nitrate, isopentyl nitrate and 2-ethylhexyl nitrate) during the reaction. The drift tube voltage was 600 V, the reactor
chamber pressure was 2.19 mbar and the drift tube temperature was 333 K. These values correspond to an E/N value of 136
Townsend (1 Townsend = $10^{-17}$ V cm$^{-2}$), where E is the electric field strength (V cm$^{-1}$) and N is the ambient air number density
within the drift tube (molecule cm$^{-3}$). Measurement were performed using the Multiple Ion Detection (MID) mode on a short
list of 11 – 13 preselected $m/z$ values resulting in measurement cycles of 25 – 35 s. This list includes the hydronium ion isotope
$H_3^{18}O^+$ (at $m/z$ = 21) and its water clusters, $H_2O \cdot H_3O^+$ and $(H_2O)_2 \cdot H_3O^+$ (at $m/z$ = 37 and $m/z$ = 55 respectively) as well as
parasitic ions $NO^+$ and $O_2^+$ ($m/z$ = 30 and $m/z$ = 32) for diagnostic purpose. The remaining ions correspond to the protonated
reference compound (methanol at $m/z$ = 33) and to the 5 – 7 major products of organic nitrates. The individual fragmentation
patterns of the organic nitrates determined during a series of preliminary measurements in scan mode (21 – 200 amu) are listed





in Table S3. All data was corrected by normalizing the ion signals with the number of hydronium ions in the drift tube, which
was calculated by multiplying the signal at $m/z = 21$ (the ionic isotope $H_3^{18}O^+$) by 500.

Isopentyl nitrate and 2-ethylhexyl nitrate were analyzed by PTR-MS for the first time. The other compounds, isopropyl nitrate,
isobutyl nitrate and 1-pentyl nitrate were previously investigated by Duncianu et al., 2017 and Aoki et al., 2007. Table S3
compiles the different fragments and relative intensities for all the studied organic nitrates in this work. Compared to the
previous studies, similar trends in the fragmentation of organic nitrates were found. Fragments with the highest relative
intensities correspond to the $R^+$ fragment, produced after losing the nitrate group in the drift tube and the $NO_2^+$ ion, which was
detected for all organic nitrates (except for 2-ethylhexyl nitrate). Other relevant fragments were the $RO^+$ fragment and/or other
ions (such as $C_3H_5^+$, $C_3H_7^+$, $C_4H_9^+$) which were formed by further fragmentation of the $R^+$ fragment.

The minimum standard sensitivity normalized to $10^6$ hydronium ions for all organic nitrates was determined prior the start of
the reaction, by calculating the maximum possible gas-phase concentration assuming the Henry's Law equilibrium (see Table
S3). The sensitivities range from 3 to 7 ncps ppbv$^{-1}$ for all organic nitrates. From these sensitivities, it was calculated that in
the reactor's headspace, they were detectable at concentrations higher than $1 \cdot 10^{-7}$ mol L$^{-1}$ in the aqueous-phase.

For methanol, assuming Henry's Law equilibrium, using a $K_H = 204$ mol L$^{-1}$ atm$^{-1}$ (average value out of those reported in
Sander, 2015), the standard sensitivity normalized to $10^6$ hydronium ions would be $9 \pm 1$ ncps ppbv$^{-1}$, corresponding to an
effective sensitivity $> 360$ cps ppbv$^{-1}$ at the typical $H_3O^+$ count rate $4 \cdot 10^7$ cps. It would be detectable in the reactor's headspace
at aqueous concentrations higher than $9 \cdot 10^{-6}$ mol L$^{-1}$.

### 2.4.2 UHPLC-UV

All the investigated organic nitrates show an intense UV absorption around 200 nm (Fig. S1). Aliquots of the solution were
sampled every 3 minutes from the reactor and the least volatile target compounds were quantified by UHPLC-UV (Thermo
Scientific Accela 600) at 200 nm. The device was equipped with a Hypersil Gold C18 column (50 x 2.1 mm) with a particle
size of 1.9 µm and an injection loop of 5 µL. A binary eluent of $H_2O$ and $CH_3CN$ was used for all analyses at a flow rate of
400 µL min$^{-1}$. Two gradients were used depending on the compounds' polarity. For isopropyl nitrate, the gradient started from
$H_2O/CH_3CN$ 80/20 (v/v) to 50/50 (v/v) for 3 min, held at this proportion for 1 minute and then set back to 80/20 (v/v) within
10 seconds until the end of the run, at minute 5 (Method A). For more polar compounds, i.e. α-nitrooxyacetone, 1-nitrooxy-2-
propanol and isosorbide 5-mononitrate, a similar gradient was employed but the initial and final proportions were $H_2O/CH_3CN$
90/10 (v/v) in order to optimize their retention times (Method B).

Calibration curves were optimized to obtain a good linearity between $5 \cdot 10^{-5}$ mol L$^{-1}$ and $1 \cdot 10^{-3}$ mol L$^{-1}$ with a $R^2 > 0.9995$.
The retention times were 0.9, 1.1, 1.2 and 2.4 min for 1-nitrooxy-2-propanol, isosorbide 5-mononitrate, α-nitrooxyacetone and
isopropyl nitrate, respectively with a limit of detection of $9 \cdot 10^{-6}$ mol L$^{-1}$ for isopropyl nitrate and $1 \cdot 10^{-5}$ mol L$^{-1}$ for the other
three compounds.





## 2.5 Reagents

Chemicals were commercially available and used as supplied: isopropyl nitrate (96%, Sigma Aldrich), isobutyl nitrate (98%, Sigma Aldrich), 2-ethylhexyl nitrate (97%, Sigma Aldrich), 1-pentyl nitrate (98%, TCI Chemicals), isopentyl nitrate (98%, TCI Chemicals), isosorbide 5-mononitrate (98%, Acros Organics), $H_2O_2$ (30%, non-stabilized, Acros Organics), $FeSO_4 \cdot 7H_2O$ (99%, Sigma Aldrich), $H_2SO_4$ (95-98%, Merck), chloroacetone (95%, Sigma Aldrich), $AgNO_3$ (99%, VWR Chemicals), KI (98%, Sigma Aldrich) and $NaBH_4$ (98%, Sigma Aldrich). Methanol (Fisher Chemical), acetonitrile (Fisher Optima) and isopropanol (Honeywell) were LC/MS grade and used as supplied. Acetone (Carlo Erba Reagents) and ether (Fisher Chemical) were HPLC grade. Tap water was purified with a Millipore MillQ system (18.2 M$\Omega$ cm and TOC < 2 ppb).

Non-commercial organic nitrates, i.e. $\alpha$-nitrooxyacetone and 1-nitrooxy-2-propanol were synthesized and purified. $\alpha$-Nitrooxyacetone was synthesized by the nucleophilic substitution reaction of iodoacetone which was synthesized previously from chloroacetone. The ketone group from $\alpha$-nitrooxyacetone was reduced to produce 1-nitrooxy-2-propanol. See S1 for further details.

## 3 Results & Discussion

### 3.1 Validation of the kinetic method

Three different kinds of validation experiments were performed. Their goals were 1) to verify that the partition of the reactants between the aqueous-phase and the gas-phase (in the reactor's headspace) was rapidly reached during the reaction; 2) to intercompare the two methods (PTR-MS and UHPLC-UV); and 3) to validate the aqueous-phase kinetic $k_{OH}$ rate constants using well known values, i.e. those of acetone and isopropanol.

Figure 2a shows the headspace signal measured by PTR-MS during the aqueous-phase ·OH-oxidation of isopropyl nitrate. Light blue background indicates periods when the solution of $Fe^{2+}$ (0.06 M) was dripped into the reactor to produce ·OH radicals. One can clearly observe how the isopropyl nitrate signal decay starts immediately at the beginning of the addition of $Fe^{2+}$ and ceases promptly when it is stopped. The partition between the aqueous-phase and the reactor's headspace is therefore swiftly reached and is faster than the PTR-MS measurement cycle rate (25 s for the shortest cycle).

Figure 2b shows the isopropyl nitrate decay during its aqueous-phase ·OH-oxidation while monitoring its concentration with the PTR-MS and the UHPLC-UV. It is clear from the figure that the kinetics are identical for both methods. Furthermore, the ratios $k_{OH,isopropryl\ nitrate}/k_{OH,methanol}$ were 0.29 ($\pm$ 0.06) and 0.31 ($\pm$ 0.06) using respectively the PTR-MS method, Eq. (4), and the UHPLC-UV one, Eq. (5). These results obtained for isopropyl nitrate confirm that its gas-phase concentrations in the reactor's headspace are proportional to its aqueous-phase concentrations. In addition, this proves that there is no ·OH attack in the gas-phase, which agrees with our previous calculation of the quantity of ·OH radicals that partitions to the reactor's headspace. Furthermore, it shows that there are no further interferences in the reaction other than the compound evaporation.





The validation of the method was achieved by determining the aqueous-phase ·OH-oxidation rate constants for acetone and isopropanol, which have been extensively studied in the literature (Table S1). Both compound decays were monitored in the reactor's headspace with the PTR-MS and their $k_{OH}$ values were determined using Eq. (2). The experiments were triplicated. The uncertainty of the experiment was calculated by the propagation of the three values standard deviation and the methanol rate constant uncertainty.

Figure 3 compares the experimentally determined $k_{OH}$ values for acetone and for isopropanol in this work with the previous values reported in the literature. The determined rate constants, $1.9\ (\pm\ 0.1)\cdot10^9$ L mol$^{-1}$ s$^{-1}$ for isopropanol, and for acetone, $1.0\ (\pm\ 0.2)\cdot10^8$ L mol$^{-1}$ s$^{-1}$, agree very well with the reported values within the experimental uncertainties.

These results show the relevance of the developed method for accurately determining the aqueous-phase ·OH-oxidation rate constant of any semi-volatile or non-volatile compound detectable either by PTR-MS or by UHPLC-UV. Hereafter, the new

rate constants for organic nitrates are presented and discussed.

### 3.2 New aqueous-phase $k_{OH}$ determinations: application of the kinetic method to organic nitrates

Aqueous-phase $k_{OH}$ for organic nitrates were determined for the first time using the developed competition method. The kinetic rate constants were calculated by Eq. (4) for isobutyl nitrate, 1-pentyl nitrate, isopentyl nitrate and 2-ethylhexyl nitrate; by Eq. (3) for α-nitrooxyacetone, 1-nitrooxy-2-propanol and isosorbide 5-mononitrate; and by Eq. (4) and Eq. (5) for isopropyl nitrate.

Figure 4 shows some examples of the linear regressions obtained for each organic nitrate where the slope corresponds to the $k_{OH,X}/k_{OH,M}$ ratio. It evidences the diversity of values obtained for the studied organic nitrates, which all fall within less than an order of magnitude from that of methanol, thus confirming that this reference is appropriate for the studied molecules.

The determined aqueous-phase ·OH-oxidation rate constant for the organic nitrates are compiled in Table 2. To account for the small number of experiments performed for some molecules, the uncertainties are given by the confidence limits of 95 %

given by the Student's t-distribution. This explains why the values obtained for 1-pentyl nitrate or 2-ethylhexyl nitrate show the largest uncertainties as their $k_{OH}$ were determined by only two experiments. Even though there are no other data available in the literature to compare with, the values obtained for $k_{OH}$ are consistent regarding the chemical structures: they reflect that the rate constant increases with the number of reactive sites.

In more details, the chemical structure of the organic nitrate and the position of the nitrate group plays an important role on

the ·OH-oxidation rate constant. Figure 5a compares these values with the rate constants of their corresponding alcohols and non-functionalized molecules. It shows how the ·OH-oxidation is reduced when the nitrate group (–ONO$_2$) is replacing an alcohol group (–OH) or a hydrogen (–H). The nitrate group attached to a primary carbon atom (i.e. isobutyl nitrate, 1-pentyl nitrate, isopentyl nitrate, α-nitrooxyacetone and 1-nitrooxy-2-propanol) slows down the reactivity by a factor of 1.7 to 3.1 in comparison to their H-substituted homologues (isobutane, pentane, isopentane, acetone and isopropanol). For isopropyl nitrate,

a secondary organic nitrate, the ·OH-attack is reduced by an order of magnitude compared to *n*-propane. 2-Ethylhexyl nitrate presents a surprisingly low reactivity towards ·OH, even lower than smaller alkyl nitrates. It is possible that even with a low





initial concentration ($5 \cdot 10^{-5}$ mol L$^{-1}$), well below its solubility threshold ($1 \cdot 10^{-4}$ mol L$^{-1}$) the complete dissolution of the compound was not achieved when the reaction started, thus inducing an underestimation of its rate constant.

A reduction of the reactivity caused by the presence of the nitrate group also occurs in the gas-phase but in a much slighter manner (Fig. 5b). It has been discussed that the nitrate group has an electron withdrawing nature that strengthens the C–H bond of the α- and the β-carbon atom thus lowering the hydrogen abstraction (Atkinson et al., 1982). However, the reduction of the reactivity is clearly more pronounced in the aqueous-phase, indicating a likely solvent kinetic effect which lowers more effectively the reaction. This could be induced by the stabilization of the reactant (Koner et al., 2007) or by the formation of a solvent barrier provoked by the nitrate group solvation which could hinder the attack of the hydroxyl radical. The latter

inhibiting effect could hinder the ·OH-attack further than the β position, up to the γ position.

In order to evaluate this question and to predict the reactivity of other organic nitrates with ·OH radicals in the aqueous-phase, we have extended the Structure-Activity Relationship method (SAR) initially built by Monod and Doussin, 2008 and Doussin and Monod, 2013 to this class of compounds.

**3.3 Structure-Activity Relationship (SAR) for ·OH-oxidation rate constants for organic nitrates**

**3.3.1 SAR Principles**

Using the experimentally determined aqueous-phase ·OH-oxidation rate constants, the aqueous-phase SAR previously developed by Monod and Doussin (2008) and Doussin and Monod (2013) was extended to organic nitrates. Briefly, the principle of the estimation assumes that the overall rate constant for the ·OH radical induced H-abstraction is equal to the sum of each kinetic rate of each reactive site. These partial kinetic rate constants are determined by considering the chemical

environment of the function along the carbon skeleton. Each –CH$_3$, –CH$_2$–, –CH<, –OH and –CHO function of the molecule is associated with a group kinetic rate constant: k(group). To consider both field and resonance effects, the rate constants k associated with each H-bearing function are modulated with both the α-neighboring effect (represented by the F parameters) and the β-neighboring effect (represented by the G parameters). This aqueous-phase SAR can predict the ·OH-oxidation rate constants of alkanes, alcohols, diols, geminal diols, carbonyls, carboxylic acids, carboxylates, and cyclic compounds as well

as polyfunctional molecules. For organic nitrates, in addition to the F(–ONO$_2$) and G(–ONO$_2$) parameters, one specific parameter, H(–ONO$_2$) was included to test the influence of the nitrate group on the reactivity on distant reactive sites in γ position, Eq. (6):

$$k = \sum_{i=1}^{n} \left( k(i) \cdot \prod F(\alpha- \text{group}) \cdot \prod G(\beta- \text{group}) \cdot \prod H_{ONO_2}(\gamma- \text{group}) \cdot C(cycle) \right), \tag{6}$$

where $k(i)$ is the partial rate constant for a –CH<, –CH$_2$–, –CH$_3$,–CHO or –OH; $\prod F(\alpha- \text{group}) \cdot G(\beta- \text{group})$ is the product

of all the contribution factors of the different neighboring groups for a reactive site in the α and β positions respectively; $H_{ONO_2}(\gamma- \text{group})$ is the contribution factor specific of the nitrate function for a reactive site in the γ position; and $C(cycle)$ is a contribution factor which represent the cycle effect strains and only affects to reactive positions inside a cycle. For detailed examples where Eq. (6) is applied to calculate the aqueous-phase k$_{OH}$ for organic nitrates can be found in S2.





In order to include in this new extended SAR a complex polyfunctional molecule such as isosorbide 5-mononitrate, which
comprises two 5-atom cycloether structures, the neighboring effects of ethers and cycloethers have also been determined, i.e.
F(-O-), G(-O-) and a parameter X(–O–) used to calculate the cycle effect strains for cycloethers by multiplying the existing
C(cycle) factor by a X(–O–) contribution factor which indicates the presence of one or two oxygen atoms in the cycle.
Furthermore, as all carbonyl compounds α-nitrooxyacetone can be hydrated in water leading to an equilibrium with its geminal
diol form, the equilibrium constant, $K_{hyd}$, is defined, if water activity is considered as unity, by Eq. (7).

$$K_{hyd} = \frac{[gem-diol]}{[carbonyl]} ,\tag{7}$$

The reactivity of these partner molecules toward ·OH radicals can be significantly different (Doussin and Monod, 2013),
however, the experimental determination only refers to the global rate constant, defined by Eq. (8).

$$k_{overall} = \frac{K_{hyd} \cdot k_{gem-diol} + k_{carbonyl}}{K_{hyd} + 1} ,\tag{8}$$

where $k_{overall}$ is the overall rate constant for the ·OH-oxidation and $k_{gem-diol}$ and $k_{carbonyl}$ the calculated rate constants for
the related species. When performing the SAR calculation, this equilibrium was considered, and descriptors were proposed to
determine both the carbonyl + ·OH and gem-diol + ·OH rate constants and the overall rate constant was calculated to be
compared with the experimental data.

Due to the absence of any experimental determination of $K_{hyd}$ in the literature for organic nitrates, a specific experiment was
performed to determine the hydration constant of α-nitrooxyacetone. This was done by dissolving the compound in $D_2O$ and
measuring the NMR ratios between the signals associated with the geminal diol configuration and the ones associated with the
carbonyl form. Details on these experiments as well as the α-nitrooxyacetone NMR spectra are presented in S3. The hydration
constant for α-nitrooxyacetone was determined to be $K_{hyd} = 0.048 \pm 0.002$ at 298 K.

### 3.3.2 Database for the extended SAR

A dataset of 24 experimental $k_{OH}$ rate constants for linear ethers and 8 for cycloethers from the literature complemented by 7
organic nitrates $k_{OH}$ rate constant determined in this work were used together to determine the contribution factors of each
group. Table S4 compiles all compounds with the corresponding values of experimental $k_{OH}$ rate constants and their associated
uncertainties. The rate constants of ·OH-oxidation of all mentioned ethers were previously determined using competition
kinetic methods. The rate constants were re-calculated considering updated values for the reference compounds. For the latter,
recommended values were chosen in most cases, however, when no recommendation was mentioned in the literature, or when
more recent studies were published, average values were calculated discarding the outliers values from the Dixon's Q test
(Table S1).



The F, G, H, and X parameters were varied simultaneously using the Microsoft® Excel® Solver routine to solve the multivariate linear regressions in order to minimize the sum of the square difference between calculated and experimental values normalized by the experimental uncertainties Eq. (9).

$$Q = \sum_i \frac{(k_{i,exp} - k_{i,sim})^2}{\sigma_i^2} , \tag{9}$$

where $k_{i,exp}$ and $k_{i,sim}$ are respectively the experimental and simulated $k_{OH}$ for compound $i$, and $\sigma_i$ is the experimental uncertainty. This target Q value allowed us to give priority, in the simulations, to experimental data determined with low uncertainties, thus improving the reliability of the SAR compared to the previous developments. The experimental uncertainties were determined thoroughly for organic nitrates (section 4.2) and were directly used for the SAR, except for 2-
ethylhexyl nitrate which was excluded from the calculations due to the presumption of slow solubilization during the kinetic experiment as mentioned. For ethers, the associated experimental uncertainties were recalculated by considering not only the linear plot uncertainties but also the uncertainties associated with the values for the reference compound, using the propagation of uncertainty. For the reported values with no uncertainty, we arbitrarily assigned a 100% uncertainty to prevent from an excessive contribution of unclear determinations. This was the case, for example, for all the determinations reported in the
works from Anbar et al., 1966 and Eibenberger, 1980.

The preexisting parameters of the SAR were not modified and they were used as reported in Doussin and Monod, (2013) and Monod and Doussin, (2008). To calculate the new parameters, in a first step, a rough estimation of the ether parameters (F(–O), G(–O–), and X(–O–)) was performed using our dataset by starting from different initial conditions. In a second step, a test of different constraints on organic nitrate parameters was performed using the ether parameters estimated in the first step (for
isosorbide 5-mononitrate). Due to the restricted database for organic nitrates, constraints were settled on the values of F(–ONO₂), G(–ONO₂) and H(–ONO₂) and five different cases were tested (Table 3) in order to find the best compromise between a minimum number of independent variables and a realistic parameterization of the possible long-range deactivating effects of the nitrate group. The five different cases were developed to investigate if the deactivating effect affects only the reactive sites in α position to the nitrate group (Case 1) or reaches the reactive sites in β position (Case 2 and Case 3) or in γ position (Case
4 and Case 5). In the final step, once the optimal case was selected, all the new parameters were simultaneously adjusted using the whole dataset (Table 4).

### 3.3.3 SAR Results

The results obtained for the 5 cases tested are shown in Table 3. Different parameters were evaluated in order to choose the
optimal case. The correlation slope between the simulated and the experimental $k_{OH}$ rate constants and their correlation coefficient, the Q values, and the relative difference between the simulated and the experimental rate constant was calculated for each organic nitrate using Eq. (10):





$$\Delta = \frac{k_{exp} - k_{sim}}{k_{exp}}, \qquad\qquad (10)$$

the sum of all the individual $\Delta$ factors is given in Table 3. A value close to 0 indicates the absence of any significant bias.

In all cases, each contribution factor for organic nitrates which was varied resulted in a value lower than 1, and the nearer to the nitrate group the lower the value was obtained, confirming the deactivating effect of the group. Nonetheless, when Cases 1, 2 and 5 were run, the calculated value for F(–ONO$_2$) dropped to 0, meaning a total suppression of the H-abstraction in the $\alpha$ position, which is unlikely, thus discarding these three cases. In the two remaining cases (3 and 4), the nitrate group impacts on the reactivity up to the $\beta$ or the $\gamma$ position, respectively. Most of the evaluation parameters are very similar in both cases,

but Case 4 is much less constrained than Case 3 which holds only one variable (Table 3). Case 3 was thus selected as the optimal case. Furthermore, the value obtained for H(–ONO$_2$) in Case 4 (0.92) is close to unity, thus showing a very slight influence of the nitrate group in the $\gamma$ position, and the difference between this value and unity potentially falls in the experimental uncertainties. The results indicate that the impact of the nitrate group on the reactivity towards $\cdot$OH radicals only affects the $\alpha$- and the $\beta$-reactive sites. This means that the reduced reactivity (observed in Fig. 5) is caused by the electron

withdrawing effect of the nitrate group which is enhanced in the aqueous-phase.

In the final step, using Case 3 constraints, all the new parameters were simultaneously adjusted using the whole dataset, and the results are listed in Table 4. The resulting neighboring factors for organic nitrates are F(–ONO$_2$) = G(–ONO$_2$) = 0.17. For ethers, the results show that F(–O–) = 1.10, a value higher than unity and G(–O–) = 0.33, lower than 1. This reveals an influence similar to the –OH group, for which F(–OH) = 2.10 and G(–OH) = 0.44. For these groups, the $\alpha$-position is activating due to

a positive mesomeric effect, whereas the $\beta$-position is deactivating due to the oxygen electron-withdrawing effect. Furthermore, the resulting value for X(–O–) = 1.79 reflects that the presence of oxygen atoms in a cycle increases the reactivity of the molecule.

Figure 6 shows the correlation between the calculated and the experimental rate constants showing a good linearity. The efficiency of the proposed extended SAR for organic nitrates, ethers and cycloethers was studied by calculating the relative

difference between the simulated and the experimental rate constant ($\Delta$ factor, see Eq. (10)). Overall, for 51 % of the experimental values, the efficiency was better than 75 % ($|\Delta| < 0.25$), and better than 60 % ($|\Delta| < 0.4$) for 69 % of experimental values. For organic nitrates, 6 out of the 7 studied molecules presented an efficiency better than 60 %. Compared to the previous versions of the SAR tested for other functional groups (Doussin and Monod, 2013 and Monod and Doussin, 2008) the efficiencies were lower. This may be due to the way the SAR parameters were calculated in this work. While the previous

SAR targeted at minimizing $\Delta$ for a maximum number of values, in this work, a good efficiency was prioritized for the experimental values with low uncertainties using Eq. (9).

Due to the restricted database used here to build the extended SAR, the –ONO$_2$ contribution factors could not elucidate the differences between the F(–ONO$_2$) and G(–ONO$_2$) contribution factors. We acknowledge that the differences between the two factors may be significant as it happens for the gas-phase reactions (Jenkin et al., 2018). In order to better assess the detailed

influence of the nitrate group to each carbon, more experimental determinations of the kinetic rate constants should be done





for the reactivity of these compounds in the aqueous-phase. However, as we have found a good agreement between the simulated values and the experimental ones, the present SAR is useful to estimate the importance of aqueous-phase ·OH-oxidation for organic nitrates in the atmosphere.

The extended SAR was used to calculate the ·OH-oxidation rate constants of several atmospherically relevant compounds,
such as hydroxy nitrates, isoprene nitrates, and terpene nitrates. The atmospheric fate of these molecules is discussed in the next section.

## 4 Atmospheric implications

In light of these results on the aqueous-phase ·OH-oxidation of organic nitrates we estimated the importance of these processes in the atmosphere. We used the developed SAR to predict the $k_{OH}$ rate constants of other atmospherically relevant organic
nitrates to evaluate if the aqueous-phase ·OH-oxidation has a significant role on their atmospheric lifetimes. Some of the evaluated organic nitrates have been detected in field campaigns (Beaver et al., 2012; Li et al., 2018); they are expected products from isoprene or monoterpenes photooxidation (Lee et al., 2014); or they are small polyfunctional nitrates that may be formed by the fragmentation of terpene nitrates or by the oxidation of alkyl nitrates (Picquet-Varrault et al., 2020; Treves and Rudich, 2003). Selecting organic nitrates potentially relevant to atmospheric chemistry, as well as those mentioned in the
literature, we listed 49 compounds that were divided into 7 categories depending on their functionalization and chemical structure: 6 alkyl nitrates, 7 hydroxy nitrates, 7 ketonitrates, 5 aldehyde nitrates, 5 nitrooxy carboxylic acids, 7 other polyfunctional nitrates containing more than one oxygenated group, and 12 terpene nitrates corresponding to highly oxidized organic nitrates formed by the oxidation of terpenes, such as α- and β-pinene, limonene and myrcene. Table S5 appends all the studied molecules and their chemical structures and properties under the scenarios studied below.
Inspired by the work of Epstein and Nizkorodov, 2012, considering various aqueous-phase scenarios from cloud/fog conditions to wet aerosols, we evaluated the atmospheric phase partition of these 49 relevant organic nitrates, and we estimated the importance of the ·OH-oxidation reaction in the aqueous-phase. Finally, the atmospheric multiphase ·OH-oxidation lifetimes of these compounds under cloud/fog conditions were calculated and compared to their gas-phase lifetimes.

### 4.1 Aqueous- and aerosol-phase partition of organic nitrates

The partition of atmospherically relevant organic nitrates between the aqueous-, the gas- and the aerosol-phases was evaluated. First, the partition between the gas- and the aqueous-phase was calculated using Eq. (11):

$$K_{gas/aq} = \frac{n_{X,gas}}{n_{X,aq}} = \frac{\rho_W}{L_{WC}K_H RT} , \qquad (11)$$

where $n_{X,gas}$ and $n_{X,aq}$ are the number of moles of compound X that are present in the gas- and aqueous-phase respectively, $\rho_W$ is the density of water in g m$^{-3}$, $L_{WC}$ is the liquid water content in g m$^{-3}$, $K_H$ is the effective Henry's Law constant in mol
L$^{-1}$ atm$^{-1}$, $R$ is the ideal gas constant 0.082 atm L mol$^{-1}$ K$^{-1}$ and $T$ the temperature in K.





Second, the partition between the gas and the aerosol-phase was calculated using Eq. (12):

$$K_{gas/aer} = \frac{n_{X,gas}}{n_{X,aer}} = \frac{\overline{M_{aer}}\gamma_X P_X^{vap}}{C_{aer}RT},$$ (12)

where $n_{X,aer}$ is the number of moles of compound X that are present in the aerosol-phase, $\overline{M_{aer}}$ is the mean organic molar mass in the aerosol-phase in g mol$^{-1}$, $\gamma_X$ the activity coefficient of a compound X, $P_X^{vap}$ its saturation vapor pressure in atm,

and $C_{aer}$ the total organic aerosol mass concentration in g m$^{-3}$. Combining Eq. (11) and Eq. (12), one obtains the fraction of any compound in each phase. The concentration of a compound in each phase depends on the values of $K_H$ and $P_X^{vap}$.

The partition was studied for two representative atmospheric conditions: i) under typical cloud/fog conditions with a $L_{WC}$ = 0.35 g m$^{-3}$ (Fig. 7a); and ii) under wet aerosol conditions with a lower $L_{WC}$ = 3 ·10$^{-5}$ g m$^{-3}$ (Fig. 7b) (Herrmann et al., 2015). The aerosol mass concentration was set to $C_{aer}$ = 1 ·10$^{-5}$ g m$^{-3}$ with a $\overline{M_{aer}}$ = 200 g mol$^{-1}$ and T = 298 K. The values of $K_H$

were calculated using the SAR developed by Raventos-Duran et al., (2010), and the values of $P_X^{vap}$ were calculated with the group contribution method by Nannoolal et al., (2004) and Nannoolal et al. (2008). Both values were taken from the GECKO-A website (http://geckoa.lisa.u-pec.fr/generateur_form.php). The activity coefficient was supposed to be 1 due to the lack of experimental or simulated data, while in a real organic aerosol, $\gamma_X$ can range from 0.8 to 10 (Wania et al., 2014). Despite this assumption, Eq. (12) is still useful to understand if the partition into the aerosol-phase is important compared to the aqueous-

phase, under the two representative conditions investigated.

Figure 7 shows the partition of the selected 49 relevant organic nitrates in the three phases. Under cloud/fog conditions ($L_{WC}$ = 0.35 g m$^{-3}$, Fig. 7a) all compounds partition between the aqueous-phase (blue) and the gas-phase (red). The presence of terpene nitrates in the aqueous-phase is highly significant: 10 out of the 12 studied molecules are present in the aqueous-phase in proportions higher than 95%. Under the same conditions, smaller functionalized nitrates (with five carbons or less) partition

in both phases depending on their functionalization. No significant partition to the aerosol-phase is observed under these conditions.

Under wet aerosol conditions ($L_{WC}$ = 3 ·10$^{-5}$ g m$^{-3}$, Fig. 7b) smaller organic nitrates are only present in the gas-phase (red). However, this is not the case for high molecular mass compounds such as terpene nitrates which equally partition between the aqueous- (blue) and the aerosol-phase (green).

These results show that depending on their reactivity, the compounds which mainly partition into the aqueous-phase may show a different atmospheric lifetime compared to the one assessed if only the gas-phase reactions are considered.

**4.2 · OH-oxidation multiphase lifetimes of organic nitrates**

The atmospheric reactivity of each of the 49 organic nitrates with ·OH radicals in the gas-phase was compared to the aqueous-phase one using W value in Eq. (13):





$\qquad W = \frac{\frac{dn_{X,gas}}{dt}}{\frac{dn_{X,aq}}{dt}} = \frac{\rho_W}{L_{WC} K_H RT} \frac{k_{OH,gas}[OH]_{gas}}{k_{OH,aq}[OH]_{aq}},$ (13)

where $k_{OH,gas}$ and $k_{OH,aq}$ are the gas-phase and aqueous-phase ·OH-oxidation rate constants respectively in mol L$^{-1}$ s$^{-1}$,

$[OH]_{gas}$ and $[OH]_{aq}$ (in mol L$^{-1}$) are the concentrations of hydroxyl radicals in the gas- and the aqueous-phase respectively.

Figure 8 compares the ·OH reactivity of organic nitrates in the aqueous- and the gas-phase for cloud/fog conditions (Fig. 8a)

and for wet aerosol conditions (Fig. 8b) described in the previous section. The ·OH radical concentrations were set to 1.4 · 10$^6$

molecules cm$^{-3}$ (2.32 · 10$^{-15}$ mol L$^{-1}$) in the gas-phase and 10$^{-14}$ mol L$^{-1}$ in the aqueous-phase (Tilgner et al., 2013). The $k_{OH,aq}$

rate constants were calculated using the extended SAR, and Eq. (8) for compounds containing carbonyl groups. Furthermore,

the $k_{OH,aq}$ for organic nitrates containing a carboxylic acid functional group considered the contribution of both the protonated

molecule and its conjugated base. For these compounds, the pH was set to 5 and 3 respectively for cloud/fog conditions and

for wet aerosol conditions. The experimental $k_{OH,gas}$ was used when available (references in Fig. 5), were complemented,

when necessary, using the gas-phase SAR developed by Jenkin et al., (2018). The GROHME method was used to obtain the

$K_{hyd}$ for compounds containing a carbonyl group (Raventos-Duran et al., 2010). Both values were taken from the GECKO-A

website (http://geckoa.lisa.u-pec.fr/generateur_form.php).

The y-axis of Fig. 8 represents the effective Henry's Law constant for each molecule. The ratio of gas- to aqueous-phase ·OH-

oxidation rate constant (in L of air · L$^{-1}$ of atmospheric water) is shown in the x-axis. Isopleths show the W values. A W value

lower than 1 (blue background) implies that more than 50% of the target compound is consumed in the aqueous-phase. The

two dashed isopleths above and below W = 1 isopleth represent the limits for aqueous and gas-phase processing exceeding

95% respectively.

Under cloud/fog conditions (Fig. 8a), most of the studied compounds present an important aqueous-phase processing. Most of

the terpene nitrates are highly functionalized and have a high molecular weight, they are almost absent in the gas-phase, and

their ·OH-oxidation is solely governed by their aqueous-phase kinetics. Only two terpene nitrates, which are first generation

reaction products of α- and β-pinene NO$_3$·-oxidation (Li et al., 2018), have a sufficiently low K$_H$ for their ·OH removal to take

place mostly in the gas-phase (≈ 93%) due to their low functionalization.

For smaller functionalized organic nitrates, the gas-phase ·OH-oxidation reaction is important. However, one can see that an

important fraction of the compounds is located between the dashed isopleths, meaning that both the gas- and the aqueous-

phase processing are relevant in their atmospheric loss towards ·OH radicals. This behavior is highly related to the

functionalization of the organic nitrate. For small ketonitrates and aldehyde nitrates, their loss is mainly taking place in the

gas-phase. With the increase of polarity, such as for hydroxy nitrates, and even more substantially for nitrooxy carboxylic

acids, the aqueous-phase ·OH-oxidation becomes more important. For organic nitrates with more than one oxidized group,

their processing in the aqueous-phase becomes the major loss process. This observation is significant for atmospheric

chemistry since most of these organic nitrates are products of isoprene photooxidation (Fisher et al., 2016).



Figure 8b shows the relative importance of aqueous-phase ·OH-oxidation under wet aerosol conditions ($L_{WC} = 3 \cdot 10^{-5}$ g m$^{-3}$). Under these conditions, the aqueous-phase ·OH-oxidation is relevant for most of the terpene nitrates. However, under these conditions these compounds should also partition to the aerosol-phase (Fig. 7). In this phase, not only the ·OH-oxidation kinetic rate constants may differ from the gas and the aqueous-phase ones but also other oxidants, such as singlet oxygen or

photosensitized carbon, can play a crucial role in the atmospheric loss of these compounds (Kaur and Anastasio, 2018; Manfrin et al., 2019). Therefore, this chemistry should be included to precisely evaluate the atmospheric loss of terpene nitrates at these conditions. For small organic nitrates, the gas-phase ·OH-oxidation reaction remains their major loss pathway. Figure 8 shows that most of the compounds for which aqueous-phase is the major loss process, the ratio $k_{OH,gas}/k_{OH,aq}$ is higher than 1, thus inducing a slower consumption in the aqueous-phase. This point was specifically investigated in Fig. 9 and Table S5.

Multiphase ·OH-oxidation lifetimes ($\tau_{OH,multiphase}$) were calculated for each organic nitrates by taking into account the ·OH reactivity both in the atmospheric aqueous-phase and gas-phase under cloud/fog conditions using Eq. (14):

$$\tau_{OH,multiphase} = \frac{1}{\Phi_{aq}k_{OH,aq}[OH]_{aq} + \Phi_{gas}k_{OH,gas}[OH]_{gas}}, \tag{14}$$

where $\Phi_{aq}$ and $\Phi_{gas}$ are respectively the fraction of each compound present in the aqueous- and in the gas-phase. Under cloud/fog conditions, the multiphase ·OH-oxidation lifetimes range from hours to several days and they are compiled in Table

S5. Their $\tau_{OH,multiphase}$ is dependent on the chemical structure of the molecules. Large molecules, such as the terpene nitrates, or compounds with reactive groups such as aldehyde nitrates have low ·OH-oxidation lifetimes which range from 6 to 27 hours. For the other organic nitrates, their lifetimes range from 1 to 28 days, being the most persistent molecules that combine more than one nitrate group or other reactivity inhibiting groups such as carboxylic groups (see Table S5).

Among the selected organic nitrates, some compounds bear their nitrate group on a tertiary carbon atom; these species may

undergo very fast hydrolysis. Their lifetimes with respect to hydrolysis range from 1 min to 8.8 h (Takeuchi and Ng, 2019). Therefore, for most of these compounds their hydrolysis will remain the main loss reaction in the atmosphere.

To assess the impact that the aqueous-phase ·OH-oxidation has on organic nitrates lifetimes, the ·OH-oxidation lifetimes relative difference was calculated ($\tau_{OH,dif}$) for each compound using Eq. (15):

$$\tau_{OH,dif}\% = \frac{\tau_{OH,multiphase} - \tau_{OH,gas-phase}}{\tau_{OH,gas-phase}} \cdot 100, \tag{15}$$

where $\tau_{OH,multiphase}$ derives from Eq. (14), and $\tau_{OH,gas-phase}$ is the ·OH-oxidation lifetime of a compound considering only the gas-phase. For molecules which barely partition into the aqueous-phase, $\tau_{OH,dif}$ value is close to zero. For molecules which partition to the aqueous-phase, a positive (respectively negative) value of $\tau_{OH,dif}$ indicates that the aqueous-phase reactivity increases (respectively decreases) the atmospheric loss of the molecule.

Figure 9 shows $\tau_{OH,dif}$ for the studied organic nitrates. For the families of compounds such as alkyl nitrates, most of the

ketonitrates and aldehyde nitrates, which barely partition into the aqueous-phase their lifetimes remain unchanged compared to the gas-phase only reactivity. Substantial atmospheric loss is expected for hydroxy nitrates, nitrooxy carboxylic acids and





organic nitrates with more than one oxidized group as in the case of some isoprene and terpene nitrated photooxidation products. Their $\tau_{OH,dif}$ range from –75 % to 155 % implying that their global ·OH-oxidation lifetimes may be either shortened or lengthened when the aqueous-phase is considered, depending on the chemical structure of the molecule. Figure 9 shows

that for most of these compounds, especially many terpene nitrates and some polyfunctionalized isoprene nitrates, their atmospheric lifetimes significantly increase (by a factor that can exceed 140%) thus showing that their atmospheric lifetimes should be even longer than what is predicted from gas-phase only reactions.

## 5 Conclusions

A new competition kinetic method was developed to accurately determine the aqueous-phase ·OH-oxidation rate constants of

low reactivity species such as organic nitrates, using a robust reference compound such as methanol. The method was applied to determine new ·OH-oxidation kinetics of 8 organic nitrates (isopropyl nitrate, isobutyl nitrate, 1-pentyl nitrate, isopentyl nitrate, 2-ethylhexyl nitrate, α-nitrooxyacetone, 1-nitrooxy-2-propanol, isosorbide 5-mononitrate), compounds for which aqueous-phase ·OH-oxidation reaction were investigated for the first time. It was found that the nitrate group provokes an important deactivation of the ·OH-attack, similar to the corresponding gas-phase reactions but enhanced by solvent kinetic

effects. A previous developed SAR method was extended to include the nitrate group allowing the prediction of aqueous-phase kinetics for these atmospherically relevant compounds. The resulting SAR parameters for the nitrate group confirmed the extreme deactivating effect of the nitrate group, up to two carbon atoms. The achieved SAR was then used to evaluate the multiphase fate of 49 organic nitrates of atmospheric relevance, including hydroxy nitrates, ketonitrates, aldehyde nitrates, nitrooxy carboxylic acids and more functionalized organic nitrates. Among these compounds, polyfunctional organic nitrates

were found to be extremely impacted by the aqueous-phase ·OH-oxidation, not only due to their very high water solubility, but also due to their reduced reactivity in this solvent. For 50% of the proposed organic nitrates for which ·OH-reaction occurs mainly in the aqueous-phase (more than 50% of the overall removal) their ·OH-oxidation lifetimes increased by 20% to 155% under cloud/fog conditions. This was even more pronounced for terpene nitrates: for 83% of them, the reactivity towards ·OH occurred mostly (> 98%) in the aqueous-phase under fog/cloud conditions, while for 60% of these terpene nitrates their

lifetimes increased by 25% to 140% compared to their gas-phase reactivity. We demonstrate that these effects are of importance under cloud/fog conditions, but also under wet aerosol conditions, especially for the terpene nitrates. These results suggest that taking into account aqueous-phase ·OH-oxidation reactivity of biogenic organic nitrates is necessary to improve the predictions of their atmospheric fates, transport and should result into a better assessment in the distribution of air pollution, SOA formation and $NO_x$ chemistry.



**APPENDIX A: Determining aqueous-phase $k_{OH}$ monitoring the headspace with the PTR-MS**

Due to the fact that the organic compounds were diluted in the aqueous-phase, headspace analyses were based on the direct proportionality between water and gas-phase concentrations of the analytes (Karl et al., 2003). Furthermore, the influence of the gas-phase $\cdot$OH-oxidation was negligible in our system. Equation (1) can be written as:

$$n\frac{[X]_{0\,(g)}}{[X]_{t\,(g)}} = \frac{k_{OH,X}}{k_{OH,M}} \cdot ln\frac{[M]_{0\,(g)}}{[M]_{t\,(g)}}, \tag{A1}$$

where $ln([X]_{0\,(g)}/[X]_{t\,(g)})$ and $ln([M]_{0\,(g)}/[M]_{t\,(g)})$ represent the gas-phase concentration relative decay of respectively the target compound and methanol. The gas-phase relative decay of both methanol and (in some cases) the target compound were monitored by the PTR-MS instrument. The PTR-MS measured quantities in counts-per-second (cps) which can be related to absolute headspace concentrations using Eq. (A2),

$$[X]_g = \frac{1}{k_{R_{rate}}t_{R_{time}}} \cdot \frac{\sum cps(X^+)}{cps(H_3O^+)}, \tag{A2}$$

where $\sum cps(X^+)$ is the sum of the counts-per-second of all the fragments which correspond to compound X; $cps(H_3O^+)$ are the counts-per-second of the hydronium ions that protonate the molecule; $k_{R_{rate}}$ is the protonation rate constant for compound X in the PTR-MS drift tube; and $t_{R_{time}}$ is the time spent by the molecule in the drift tube. Combining Eq. (A1) and Eq. (A2), one obtains Eq. (2).

*Author contributions.* JMGS developed the experimental kinetic method. NB provided the UV-VIS data for organic nitrates. JMGS and SR developed the UHPLC-UV method for organic nitrates. JM and BTR developed the PTR-MS method for organic nitrates. JLC performed the organic nitrates synthesis and the NMR experiments to obtain the α-nitrooxyacetone $K_{hyd}$. JMGS and AM built the SAR extension to include nitrate and ether contribution factors. NB and JW significantly contributed to the SAR construction. CMV wrote the code in the GECKO-A model to apply the extended SAR to any molecule. AM and JLC
lead the project. JMGS and AM wrote the article with inputs from all coauthors.

*Data availability.* PTR-MS and UHPLC-UV raw data from the experimental method are available at https://drive.google.com/drive/folders/1cxPdk58rnGZwkzmGGetWe4AGnww356RB?usp=sharing (last access: 9 July 2020). Data related to the extended SAR or the atmospheric implications sections can be requested to JMGS
(juanmiguelgs93@gmail.com) or to AM (anne-monod@univ-amu.fr).

*Competing interests.* The authors declare that they have no conflict of interest.



*Acknowledgements.* This project has received funding from the European Union's Horizon 2020 research and innovation
programme under the Marie Skłodowska-Curie grant agreement No713750. It has been carried out with the financial support
of the Regional Council of Provence- Alpes-Côte d'Azur and with the financial support of the A*MIDEX (n° ANR- 11-IDEX-
0001-02), funded by the Investissements d'Avenir project funded by the French Government, managed by the French National
Research Agency (ANR). This study also received funding from the French CNRS-LEFE-CHAT (Programme National-Les
Enveloppes Fluides et l'Environnement-Chimie Atmosphérique – Project "MULTINITRATES"), and from the program ANR-
PRCI (ANR-18-CE92-0038-02) – Project "PARAMOUNT". Finally, the authors thank Richard Valorso for his help using the
GECKO-A modelling tool.

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



| Name | Structure | $K_{H, 298 K}$ / mol L$^{-1}$ atm$^{-1}$ | Solubility at 298 K / mol L$^{-1}$ | Monitored by |
|---|---|---|---|---|
| Isopropyl nitrate | | 0.75 | $2.5 \cdot 10^{-2}$ | PTR-MS UHPLC-UV |
| Isobutyl nitrate | | 0.60 | $9.4 \cdot 10^{-3}$ | PTR-MS |
| 1-Pentyl nitrate | | 0.74 | $2.7 \cdot 10^{-3}$ | PTR-MS |
| Isopentyl nitrate | | 0.40 | $2.5 \cdot 10^{-3}$ | PTR-MS |
| 2-Ethylhexyl nitrate | | 0.18 | $1 \cdot 10^{-4}$ | PTR-MS |
| α-Nitrooxyacetone | | $1.0 \cdot 10^{3}$ | 1.6 | UHPLC-UV |
| 1-Nitrooxy-2-propanol | | $6.7 \cdot 10^{4}$ | 0.6 | UHPLC-UV |
| Isosorbide 5-mononitrate | | $1.3 \cdot 10^{7}$ | 0.3 | UHPLC-UV |
| Isopropanol | | 130 | 16.6 | PTR-MS |
| Acetone | | 30 | 17.2 | PTR-MS |

**Table 1: Chemical structures and properties of the studied organic nitrates. Henry's Law constants were obtained from Sander, 2015 except for 2-ethylhexyl nitrate and isosorbide 5-mononitrate which were calculated using the SAR developed by Raventos-Duran et al., (2010). Organic nitrates solubilities were calculated using the model WSKOWWIN$^{TM}$.**



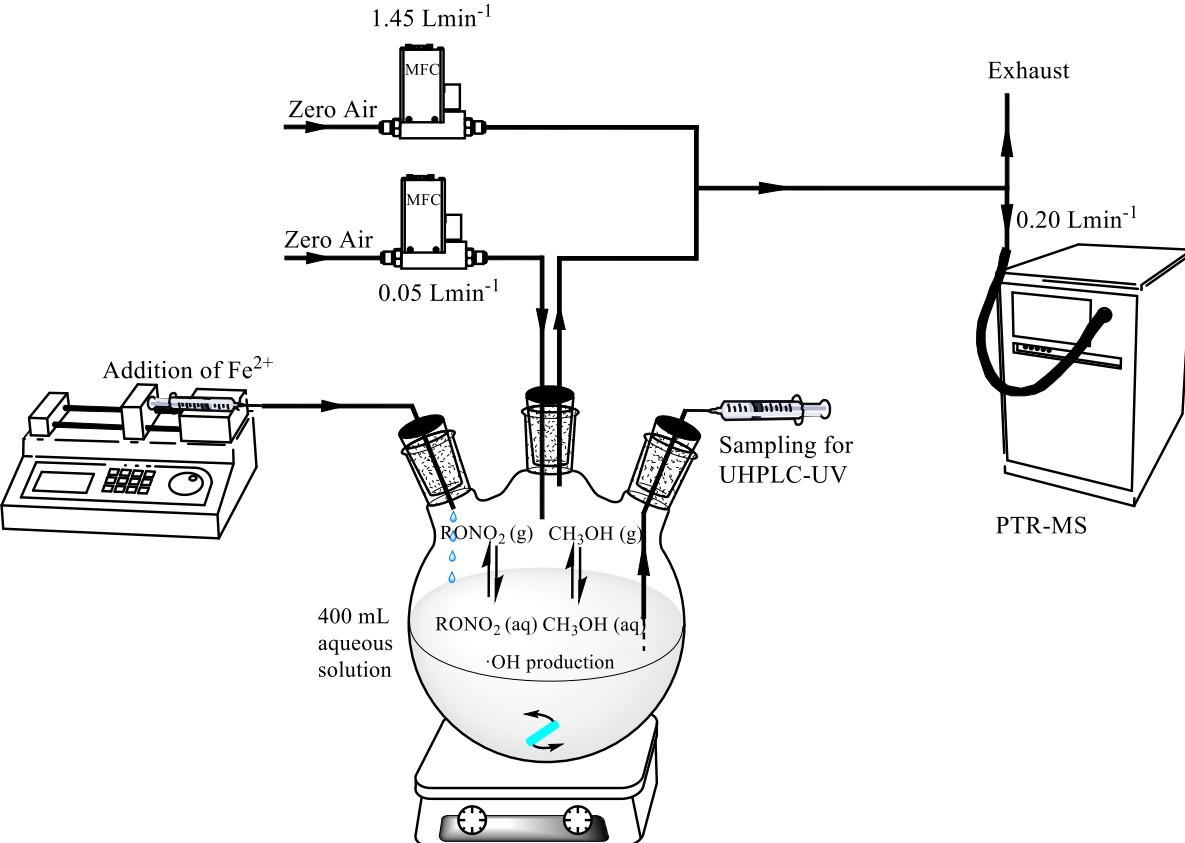

**Figure 1: Experimental set-up for the competition kinetics method**

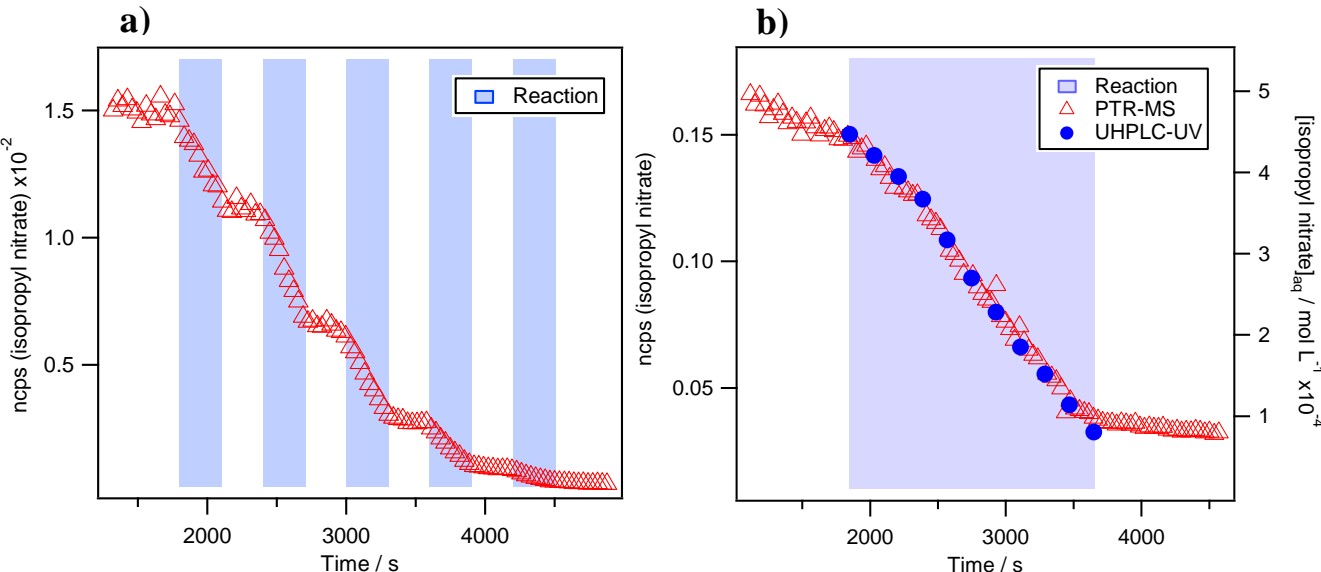


**Figure 2: Kinetic decays of isopropyl nitrate during its aqueous-phase ·OH-oxidation a) Monitored by PTR-MS during several sequential additions of the $Fe^{2+}$ solution. Initial conditions in the reactor were [Isopropyl nitrate]$_0$ = 5 ·10$^{-5}$ mol L$^{-1}$, [H$_2$O$_2$]$_0$ = 4 ·10$^{-3}$ mol L$^{-1}$ and [H$_2$SO$_4$]$_0$ = 5 ·10$^{-3}$ mol L$^{-1}$; b) Intercomparison between PTR-MS and UHPLC-UV detection. Initial conditions were [Isopropyl nitrate]$_0$ = 6 ·10$^{-4}$ mol L$^{-1}$, [CH$_3$OH]$_0$ = 3 ·10$^{-4}$ mol L$^{-1}$, [H$_2$O$_2$]$_0$ = 4 ·10$^{-3}$ mol L$^{-1}$ and [H$_2$SO$_4$]$_0$ = 5 ·10$^{-3}$ mol L$^{-1}$. In both**
**graphs the time is set to 0 when isopropyl nitrate is injected. Blue background indicates periods when the solution of $Fe^{2+}$ (0.06 M) was dripped into the reactor to produce ·OH radicals.**





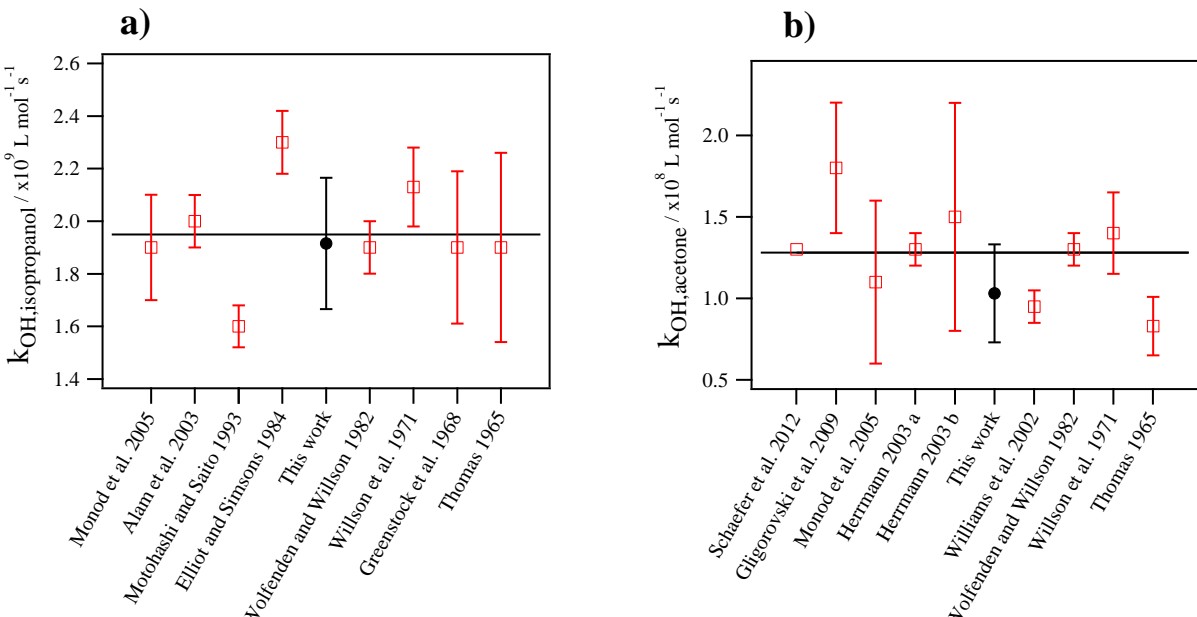

**Figure 3: Validation experiments: determination of the aqueous-phase $k_{OH}$ values using the new developed method, and comparison with the reported values in the literature for a) isopropanol and b) acetone. The horizontal line represents the average value of the previous studies.**



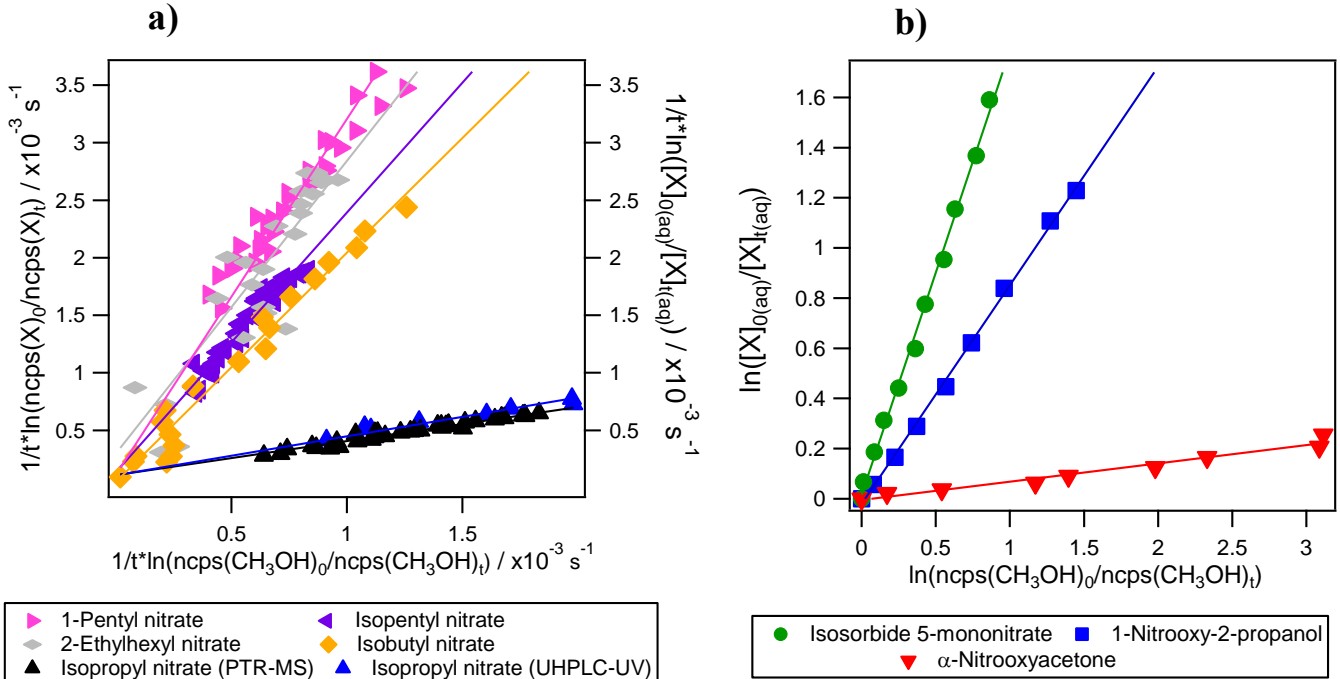

**Figure 4:** Examples of competition kinetic results for the studied organic nitrates using the new developed method: experiments 8, 14, 16, 19, 21, 24, 26 and 31 (listed in Table S2). In panel a) molecules were monitored in the reactor's headspace by PTR-MS (left axis and horizontal axis) except for isopropyl nitrate which was also monitored by UHPLC-UV (right axis). In panel b) methanol was monitored in the reactor's headspace by PTR-MS (horizontal axis) while the polyfunctional organic nitrates were followed by UHPLC-UV (left axis).



| Studied compounds | $k_{OH\ exp}$ / $\cdot 10^8$ L mol$^{-1}$ s$^{-1}$ | $k_{OH\ sim}$ / $\cdot 10^8$ L mol$^{-1}$ s$^{-1}$ | Δ |
|---|---|---|---|
| Isopropyl nitrate | 2.8 (± 0.6) | 3.0 | –0.07 |
| Isobutyl nitrate | 17 (± 11) | 13.5 | +0.22 |
| 1-Pentyl nitrate | 31 (± 46) | 31.8 | –0.02 |
| Isopentyl nitrate | 22 (± 9) | 24.6 | –0.10 |
| 2-Ethylhexyl nitrate | 24 (± 34) | 58.9 | –1.46 |
| α-Nitrooxyacetone | 0.8 (± 0.4) | 1.3 | –0.69 |
| 1-Nitrooxy-2-propanol | 8.7 (± 1.9) | 6.3 | +0.28 |
| Isosorbide 5-mononitrate | 18 (± 5) | 14.2 | +0.19 |


**Table 2: Aqueous-phase ·OH-oxidation rate constants for eight organic nitrates: comparison between the experimental rate constants ($k_{OH\ exp}$) and the simulated ones ($k_{OH\ sim}$) using the extended SAR. Calculations use the SAR parameters indicated in Table 4. The factor Δ is the relative difference between the simulated and the experimental rate constant.**






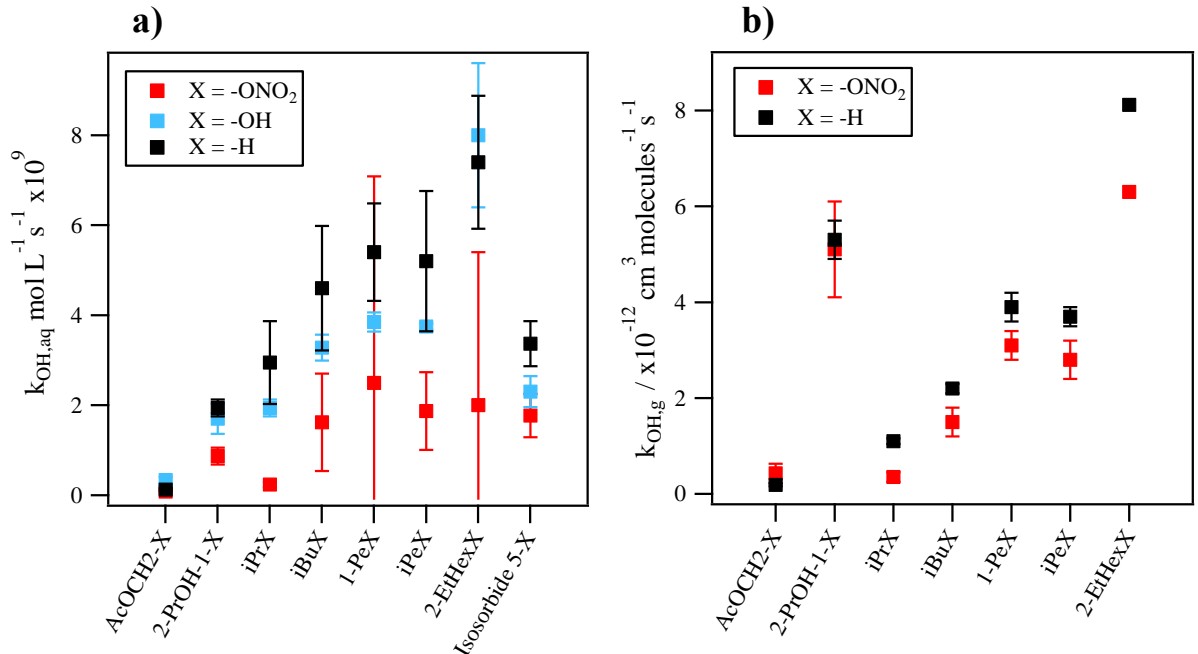

**Figure 5:** Comparison between the $k_{OH}$ values for organic nitrates, the corresponding alcohols, and non-substituted homologue compounds a) in the aqueous-phase and b) in the gas-phase. Aqueous-phase $k_{OH}$ values were taken from : Reuvers et al., (1973) and Adams et al., (1965) for alcohols; Getoff, (1991) and Rudakov et al., (1981) for H-substituted compounds; the predictions using the
SAR for hydroxy acetone and for the –OH and –H substituted homologues of 2-ethylhexyl nitrate and isosorbide 5-mononitrate. Gas-phase $k_{OH}$ values were taken from Atkinson et al., (1982); Atkinson and Aschmann, (1989); Becker and Wirtz, (1989); Bedjanian et al., (2017); Suarez-Bertoa et al., (2012); Talukdar et al., (1997); Treves and Rudich, (2003) and Zhu et al., (1991) for organic nitrates; from Atkinson et al., (1997), Atkinson et al. (1992); and Atkinson, (2003) for H-substituted homologues; and from the predictions of the SAR by Jenkin et al., (2018) for 2-ethylhexyl nitrate and its H-substituted homologue.




|  | Case 1 | Case 2 | Case 3 | Case 4 | Case 5 |
|---|---|---|---|---|---|
| **F(–ONO₂)** | 0 | 0 | **0.17** | 0.18 | 0 |
| **G(–ONO₂)** | $1^*$ | 0.33 | **0.17** | 0.18 | 0.37 |
| **H(–ONO₂)** | $1^*$ | $1^*$ | $1^*$ | 0.92 | 0.37 |
| **Slope** | 1.18 | 1.18 | **1.01** | 0.98 | 0.53 |
| **$R^2$** | 0.92 | 0.97 | **0.96** | 0.95 | 0.94 |
| **Δ** | -5.19 | -0.14 | **-0.19** | 0.05 | 2.06 |
| **Q** | 23.0 | 0.12 | **0.59** | 0.58 | 0.93 |
| **Efficiency 75%** | 29 % | 86 % | **71 %** | 57 % | 14 % |
| **Efficiency 60%** | 43 % | 100 % | **86 %** | 86 % | 43 % |

Table 3: SAR tests on organic nitrate parameters: 5 case tests investigated to evaluate the nitrate group influences. Constraints imposed: Values fixed to 1 (i.e. no influence on the reactivity) marked by *; F(–ONO₂)=G(–ONO₂) for Case 3 and Case 4; and G(–ONO₂)=H(–ONO₂) for Case 5. In bold is the selected case. Slope is the correlation slope between the simulated and the experimental $k_{OH}$ rate constants and $R^2$ is the corresponding correlation coefficient. The Δ factor is the relative difference between the simulated and the experimental rate constant. Q is the sum of the square difference between calculated and experimental values normalized by the experimental uncertainties. Efficiencies at 75% and 60% represent the percentage of organic nitrates with |Δ| < 0.25 and |Δ| < 0.40 respectively.




| Parameter | Value |
|:---:|:---:|
| F(–O–) | 1.10 |
| G(–O–) | 0.33 |
| C'5(cycle) | 1.43 |
| C'6(cycle) | 1.79 |
| F(–ONO$_2$) | 0.17 |
| G(–ONO$_2$) | 0.17 |

**Table 4: SAR results: new calculated neighboring effects parameters for ethers, cyclic ethers and organic nitrates (using Case 3 constraints).**






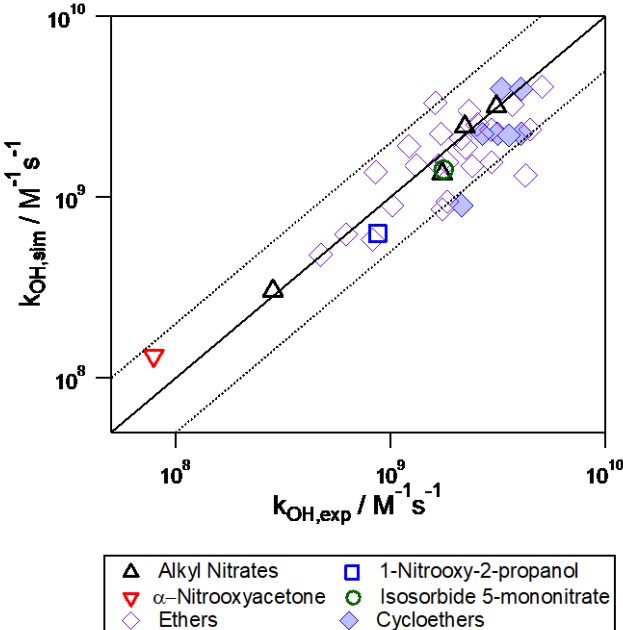

**Figure 6: Correlation plot between the experimental and the simulated aqueous-phase ·OH-oxidation rate constants for organic nitrates and ethers. The plain line corresponds to the 1:1 regression and the dashed lines represent the limits where the simulated rate constants deviate from the experimental ones by a factor of 2.**




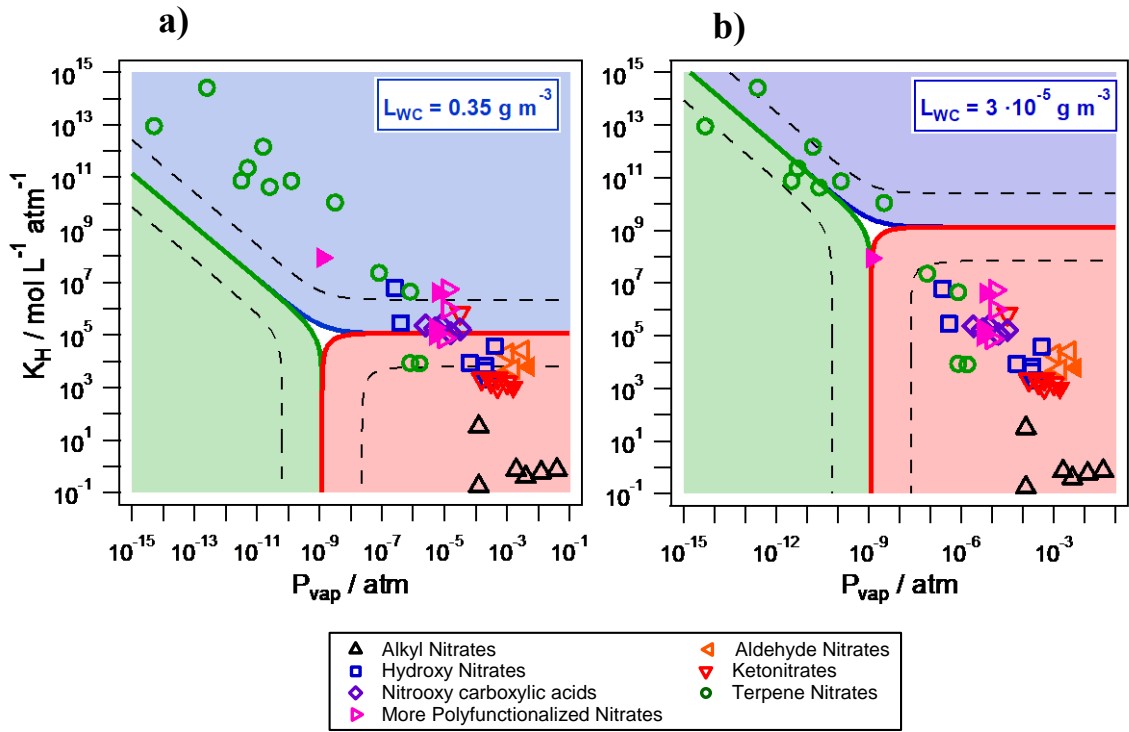

**Figure 7: Partition of atmospherically relevant organic nitrates in the aqueous- (blue), the gas- (red) and the aerosol-phase (green) for two different air parcels with a) $L_{WC}$ = 0.35 g m$^{-3}$ (cloud/fog conditions) and b) $L_{WC}$ = 3.5 ·10$^{-5}$ g m$^{-3}$ (wet aerosol conditions). Temperature was set to 298 K, the aerosol mass concentration to $C_{aer}$ = 1 ·10$^{-5}$ g m$^{-3}$ with a $\overline{M_{aer}}$ = 200 g mol$^{-1}$. The dashed lines 815 represent the limits where the concentration of one compound exceeds 95 % in each phase. Filled markers represent organic nitrates derived from isoprene.**



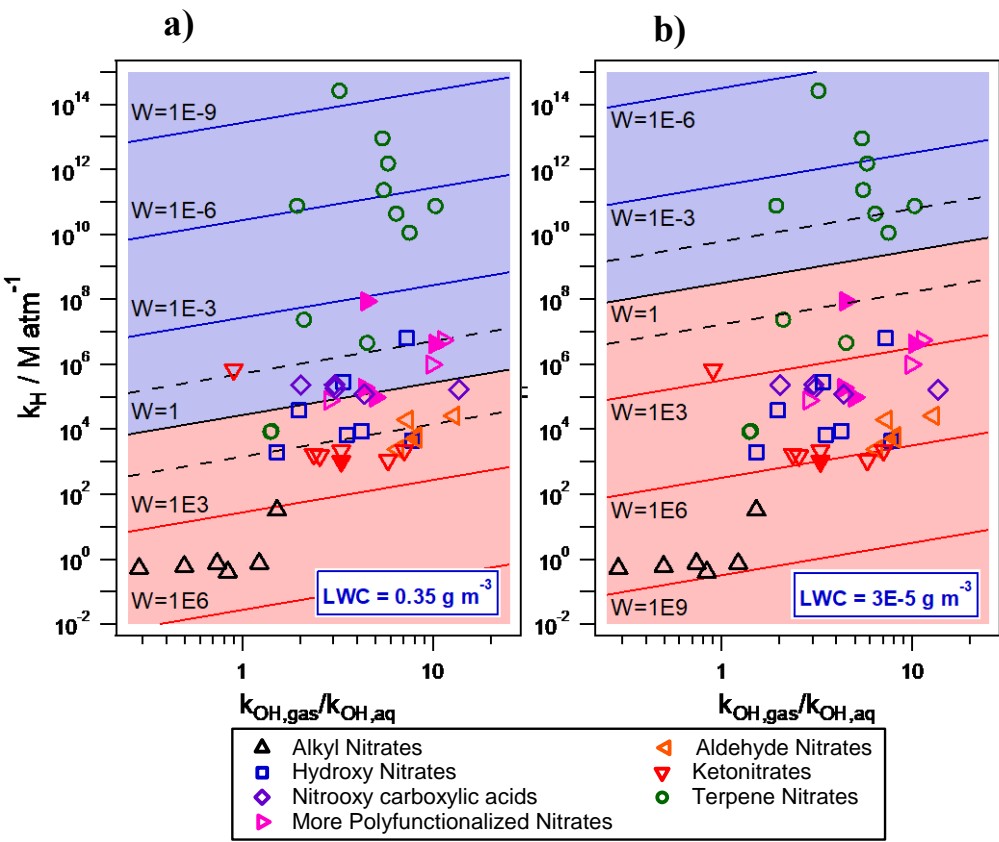

**Figure 8: Gas-aqueous partition and reactivity (towards ·OH-oxidation) of 49 organic nitrates for two different air parcels with a) $L_{WC}$ = 0.35 g m⁻³ (cloud/fog conditions) and b) $L_{WC}$ =3.5 ·10⁻⁵ g m⁻³ (wet aerosol conditions) at 298 K. ·OH radical concentrations were set to 2.32 ·10⁻¹⁵ mol L⁻¹ (1.4 ·10⁶ molecules cm⁻³) in the gas-phase and to 10⁻¹⁴ mol L⁻¹ in the aqueous-phase. Filled markers represent organic nitrates derived from isoprene.**





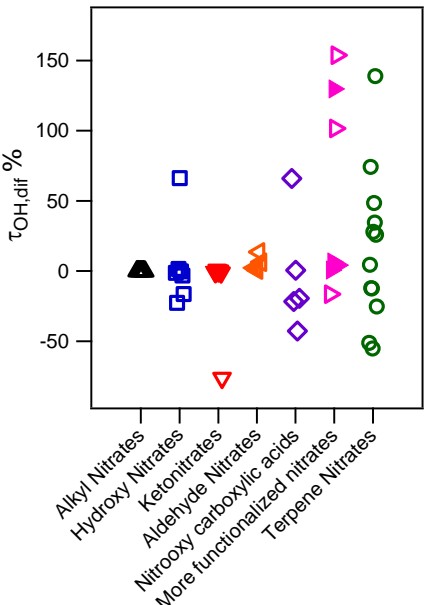

**Figure 9: Relative difference of ·OH-oxidation lifetimes between an air parcel with and without liquid water (L$_{WC}$ = 0.35 g m$^{-3}$) for atmospherically relevant organic nitrates. Filled markers represent organic nitrates derived from isoprene.**