# Peer review of "On the importance of atmospheric loss of organic nitrates by aqueousphase ·OH-oxidation"

_Atmospheric Chemistry and Physics, 2020_

## Referee Comment (RC1) · Anonymous Referee #1 · 23 Nov 2020

This manuscript prepared by Gonzalez-Sanchez represents a full-bodied investigation into the aqueous-phase reactivities of organic nitrates of atmospheric relevance. The manuscript contains a few parts: 1) experimental determination of the OH rate coefficients of eight organic nitrates 2) extension to an existing structure-activity relationship (SAR) to incorporate ethers and nitrate, and 3) estimation of the OH reactivity of 49 organic nitrates based on their new SAR and discussions on their atmospheric fate. Multiple valuable results were obtained from this study, with a major one being that the -ONO2 group exhibits a significant suppression effect on OH reactivities in the aqueous phase, to a more extent than in the gas-phase. Thus, aqueous-phase OH oxidation can lead to a longer OH oxidation lifetime for some of the organic nitrates. With emissions

of NOx (e.g., from cars) constantly reduced in many parts of the world, the formation of organic nitrates is becoming more important in the atmosphere. Organic nitrates may represent a reservoir of NOx, and their atmospheric lifetimes need a careful assessment. A reliable SAR is highly valuable for the atmospheric chemistry community, given that experimental investigation for all the species is infeasible. The topic of this work is highly important, timely, and within the scope of ACP. I strongly recommend publication in ACP after the following points are addressed:

Major comment

My only major comment is regarding the ability of the SAR to be extrapolated to a wider spectrum of compounds. The linear relationship shown in Fig 6 is between the SAR-simulated values and the experimental values from which the SAR was derived; correct? For other empirically-derived relationships, such as free energy linear relationships (LFERs), it is important to derive the relationship with a training set, followed by validation using an experimental set. With the limited number of organic nitrates, this may be challenging. What about ethers? Either way, the authors should comment on the reliability of the SAR when used for other compounds.

Minor comments

- Can any of the secondary organic nitrates used in the experiment undergo hydrolysis within the time scale of the experiment? Maybe this is confirmed with the H2O2 control experiment?

- My understanding is that Figure 7 represents the phase partitioning of these species in equilibrium. Two minor questions:

Is partitioning equilibrium achieved in the atmosphere? Maybe yes for cloudwater, how about aerosol liquid water (e.g., viscosity)

Can some of the organic nitrate be surface active and partition to the air-water interface?

- Line 466, how did the authors consider the reactivities of carboxylic acid and carboxylate? Is this parameterized in the previous version of SAR?

Technical Comments

- Line 283, the sentence, "indicating a likely solvent kinetic effect which lowers more effectively the reaction." is awkward. Maybe consider rephrasing to "implying a solvent kinetic effect that can more effectively suppress the reactivity"?

- In a few places "prior" should be "prior to" E.g., "Prior to each experiment." Check lines 114, 154, 161, 193

---

## Referee Comment (RC2) · Anonymous Referee #2 · 30 Nov 2020

The manuscript describes a new relative rate approach to measure aqueous kinetics and its application to determine rate constants for hydroxyl radical (OH) with a series of organic nitrates. The new approach is clever and takes advantage of the fact that first-order kinetics do not require knowledge of absolute concentrations of analytes, only the relative concentrations. The method is also clever in that it can couple a gas-phase reference measurement in conjunction with an aqueous-phase target measurement for compounds that have low vapor pressures. The resulting rate constants for organic nitrates are useful, as is the new structure-activity relationship (SAR) that was constructed. Finally, the paper also examines the environmental relevance of the kinetic data, which is a helpful scaling-up of the results. While I have a number of issues that

need to be addressed, overall, I am supportive of the manuscript.

»Major points«

The manuscript text is overall fine, but would be strengthened if unnecessary portions were removed (e.g., the discussion of the "direct method" to determine rate constants on lines 60 - 63), repetition was eliminated (e.g., lines 291-292 repeat lines 286-288), and the text was edited for issues such as noun-verb agreement (e.g., "Measurement were performed..." on line 178). In addition, using parentheses to try to put the opposite case inside of a sentence makes for difficult reading, e.g., line 516: "For molecules which partition to the aqueous-phase, a positive (respectively negative) value of ðÎIJŘðÌŚĆðÌŘż,ðÍŚŚðÍŚŰðÍŚŞ indicates that the aqueous-phase reactivity increases (respectively decreases) the atmospheric loss of the molecule." This should be revised.

There are some issues with pH. First, the solution is listed as "pH < 3" (line 109), but the value should be closer to pH 2 since solutions contained 5 mM H2SO4 (line 764). This should be clarified. Second, I appreciate that an acidic solution is needed to keep Fe dissolved, but pH 2 is not relevant for most atmospheric waters, which are much less acidic. The low pH values used in the experiments are a bit concerning since past work has shown that the OH rate constant is sometimes pH dependent for classes of compounds that do not have a change in acid-base speciation across the pH range (e.g., Kroflic et al., PCCP, DOI: 10.1039/c9cp05533a). Third, the SAR lumps the current rate constants with 32 literature values without any tabulation or discussion of the pH values for the literature results. It's likely they were not done at pH 2, so it's not clear they can be considered along with the organic nitrates studied in this work. This should be examined and discussed.

The 2-ethylhexyl nitrate kinetic data looks suspicious (Figure 4) and produce a rate constant that is lower than expected and not included in the SAR (line 345). The authors (line 278) suggest that the experiments might have suffered from solubility issues,

but they don't test this. As it currently stands, this compound is thus in publication purgatory: the data is reported in the manuscript, but is not trusted. The authors need to take a stand on the compound, either (1) eliminate it entirely from the manuscript or (2) do additional experiments (e.g., at a lower concentration) to try to get good data.

The calculation and discussion of the lifetimes of the organic nitrates should be improved. (a) Fates other than OH (gas and aqueous) need to be better described. There is little discussion of hydrolysis, mostly just three lines about the rapid hydrolysis of compounds with the nitrate group on a tertiary carbon (lines 506-510). But don't other nitrates also have significant hydrolysis rate constants? This should be described and quantified. In addition, isn't photolysis (in both phases) a significant sink for some of the nitrates with an aldehyde or ketone group? This should be discussed, especially in terms of how these other fates would alter the lifetimes that are presented. (b) The aqueous (and thus multiphase) lifetimes depend on the aqueous OH concentrations. The aqueous [OH] used in the manuscript (1E-14 M) is from a numerical model and is approximately 2 – 10 times higher than measurements. For example, Kaur et al. (2018) report aqueous [OH] of approximately 5E-15M in cloud drops and an extrapolated 1E-15 M in particle water. These model-measured differences, and their impact on calculated multiphase lifetimes, should be discussed. (c) On first read, Figure 8 suggests that compounds with high W values (i.e., where most oxidation occurs in the aqueous phase) will have shorter lifetimes because of this aqueous sink. But this is not the case - for many of the compounds there is very little difference between gas-phase lifetime and multiphase (i.e., gas and PM) lifetime, as discussed only later with Figure 9. It would be helpful to incorporate the lifetimes into Figure 8, e.g., as symbol color or size, to more clearly show this. Showing the lifetimes in a figure would be very helpful, as this information is currently buried in Supplemental Table S5.

»Other points«

Line 39. Hydrolysis to form nitrate is also an important sink for some nitrates, especially some isoprene-derived organic nitrates. Photolysis should be mentioned also.

l. 79. The absolute uncertainty on the k(OH+SCN-) rate constant might be larger than the target rate constant, but this doesn't matter: it is the relative – not absolute – uncertainty (i.e., 0.20/1.12) that is propagated in the relative rate calculation to determine the uncertainty on the target compound.

l. 125. To avoid any confusion that k(OH,X) is a pseudo-first order rate constant for loss of X, the authors should explicitly state that this is the second-order rate constant and give the units (M-1 s-1).

l. 135. The method works as long as the reference and target compounds are in Henry's law equilibrium in the reaction flask. For highly reactive compounds this condition might break down. Based on an estimated characteristic time for aqueous mixing in the flask, is there an upper limit to the rate constants that can be determined under their [OH] conditions?

l. 243. Where is the "...previous calculation of the quantity of OH radicals..." that is referenced here?

Section 3.2 Indicate how uncertainties were determined. Were they based on the standard deviation between replicates propagated with the uncertainty in the MeOH rate constant?

l. 429 (and throughout). Typically, the liquid water content abbreviation "LWC" has no subscripted portion.

l. 461. The gas-phase units are shown here but not described until later. Since they're not standard, clarify here that [OH](gas) is in mol-OH / L-air and that the same volume unit is used for gas-phase rate constants.

l. 471. Both values of what?

Table 2. Include the number of experiments for each compound in the table. Also indicate in the table that uncertainties are 95% CI.

Figure 5 is a helpful comparison of data across compounds and phases.

Figure 8. Put the symbol for the isoprene-derived nitrates in the figure legend (along with the rest of the classes) rather than as a sentence in the caption.

Figure 9. Include another panel that shows the analogous results for the cloud LWC condition.

The SI does not have a list of references.

Table S5 includes the multiphase lifetime for the particle LWC condition, but not for the cloud LWC condition. The latter should be included.

---

## Author Comment (AC1) · 16 Jan 2021

The authors thank the reviewers for their positive feedbacks on our work and their helpful comments and questions. We have addressed all comments and corrections as detailed below, and we modified the manuscript accordingly.

**REVIEWER 1**

**Major comment**

**My only major comment is regarding the ability of the SAR to be extrapolated to a wider spectrum of compounds.**

Although a narrow number of experimental data were used to build this SAR, we carefully chose monofunctional compounds and polyfunctional compounds in the extrapolation bearing the same chemical functions. Further arguments are also developed below.

**The linear relationship shown in Fig 6 is between the SAR-simulated values and the experimental values from which the SAR was derived; correct?**

Figure 6 plots all experimental $k_{OH}$ values against the simulated $k_{OH}$ values for the compounds which were used for the SAR construction. The data are compared to the 1:1 regression (plain line). The linear regression between the two sets is not shown for clarity.

For more clarity, we replaced the sentence in line 396 "Figure 6 shows the correlation between the calculated and the experimental rate constants showing a good linearity" by "Figure 6 shows the correlation between the calculated versus the experimental $k_{OH}$ values, and it is compared to the 1:1 regression line."

**For other empirically-derived relationships, such as free energy linear relationships (LFERs), it is important to derive the relationship with a training set, followed by validation using an experimental set. With the limited number of organic nitrates, this may be challenging. What about ethers? Either way, the authors should comment on the reliability of the SAR when used for other compounds.**

The SAR was firstly developed by Monod and Doussin (2008) for alkanes, cyclic compounds, alcohols, carboxylic acids and carboxylates and was then developed for ketones and aldehydes by Doussin and Monod (2013). We have extended the SAR to include organic nitrates and ethers by adding 4 parameters to the SAR. Those new parameters were varied while all the previous parameters remained unchanged.

For ethers, 34 $k_{OH}$ experimental values (including 8 cyclic ethers) were used to resolve 3 different variables (F(–O–), G(–O–) and X(–O–)). In the case of RONO$_2$, the dataset was more limited: 7 $k_{OH}$ values were used to resolve one variable (F(–ONO$_2$)= G(–ONO$_2$)). For this type of relationships, it is usually considered that a ratio of at least 10 experimental values for 1 variable is a rule to follow to avoid overfitting the SAR (Peduzzi et al., 1996). The approach to use a training set and then validate the results with a different set may be done only with a sufficiently large dataset. Therefore, we decided to use all the available experimental data to have the 'best possible' training set. Meanwhile, there are some aspects in our approach and in the results that can provide some validation:

1) One aspect concerns the validation of preexisting SAR parameters. Only four new parameters were added in this extended SAR, while the preexisting 26 parameters remained unchanged. Considering that the extension of the SAR with four additional parameters did not require any optimization of the preexisting parameters, we can consider that the new experimental data act as validation of these preexisting parameters.

2) This SAR extension incorporates, in respect to the previous developed SAR, the weighting of the experimental values with their uncertainties. A $k_{OH}$ value with a lower uncertainty was thus more constraining than values with higher (or no) uncertainties. This improvement of the approach renders

difficult the sorting of data to build a 'training set' and a 'validation set'. On one hand, not using the data with low uncertainties for the SAR construction may induce significant deviations in the resulting SAR parameters. On the other hand, using data with high uncertainties for the validation cannot be completely trusted. In our approach, all experimental values were included in the training set, thus data bearing low uncertainties had a higher influence on the determined parameters. The fact that only a few significant discrepancies (deviations by a factor of 2) between the experimental and simulated values even for low-weighted experimental values gives some validation of the final parameters.

3) Furthermore, the facts that the resolved parameters were chemically coherent, and that a wide range of reactivities was covered (two orders of magnitude), provide evidence to rely on the application of the extended SAR to other compounds.

4) A third aspect concerns isosorbide 5-mononitrate that can provide some validation of our SAR: this molecule presents a highly complex polyfunctional structure, with 2 ethers, 2 cycles, one alcohol and one nitrate group. The fact that its simulated $k_{OH}$ value (14.2 x $10^8$ L mol$^{-1}$ s$^{-1}$) agrees (within uncertainties) with the experimental value (18 ($\pm$5) x $10^8$ L mol$^{-1}$ s$^{-1}$) gives some validation of the extended SAR application to polyfunctional compounds.

We have modified the manuscript highlighting that i) the good results obtained even for the low-weighted values, ii) the chemical coherence of the resulting SAR parameters, iii) the large range of $k_{OH}$ values covered by the data, and that iv) some validation can be done using isosorbide 5-mononitrate.

In line 397: "It shows a good linearity covering a wide range of reactivities (two orders of magnitude)."

In line 405: "Several arguments provide some evidence on the reliability of the application of the SAR to other compounds: i) there are only a few discrepancies (deviations by a factor of 2) between the experimental and simulated values, even for low-weighted experimental values, ii) the resolved parameters were chemically coherent, iii) a wide range of reactivities was covered (two orders of magnitude) and iv) isosorbide 5-mononitrate presents a highly complex polyfunctional structure (two cycles, two ether, one alcohol and one nitrate groups), the good agreement between its simulated and experimental $k_{OH}$ values (within uncertainties, Table 2) gives some validation of the extended SAR application to polyfunctional compounds."

**Minor comments**

**Can any of the secondary organic nitrates used in the experiment undergo hydrolysis within the time scale of the experiment? Maybe this is confirmed with the H2O2 control experiment?**

In addition to the $H_2O_2$ control experiments, control experiments were performed at pH = 2–3 for all organic nitrates within the time scale of the experiments. None of them showed any decay due to reaction with $H_2O_2$ or hydrolysis of the nitrate group, and we verified that none of them showed any formation of $NO_3^-$, product of organic nitrates hydrolysis (Darer et al., 2011) .

We added in the manuscript, in line 115: "Furthermore, control experiments at pH = 2.5 confirmed that none of the organic nitrates underwent hydrolysis at significant rates under our experimental conditions."

**My understanding is that Figure 7 represents the phase partitioning of these species in equilibrium. Two minor questions:**

**Is partitioning equilibrium achieved in the atmosphere? Maybe yes for cloudwater, how about aerosol liquid water (e.g., viscosity)**

We agree with Reviewer 1 on that point: in their kinetic flux model Shiraiwa and Seinfeld (2012) demonstrate that equilibrium is achieved in the order of seconds or minutes for the phase partitioning of relatively high volatility organic compounds into liquid particles, while the equilibration timescale

ranges from hours to days for semi-solid viscous particles, low volatility species or large particle sizes, and thus can be subject to equilibrium shifts by faster chemical reactions in each phase. However, these considerations are beyond the scope of our manuscript. Our goal is to show how aqueous-phase chemistry can affect organic nitrate fate even at very low LWC. To do so, we consider a very simple model of deliquescent particles containing a LWC of $3.5 \cdot 10^{-5}$ g m$^{-3}$, and assuming fast phase partitioning.

We have added a sentence explaining this assumption in the manuscript:

In line 458: "It is worth noting that in the atmosphere, the equilibrium may not be instantaneous. In their kinetic flux model, Shiraiwa and Seinfeld (2012) demonstrate that equilibrium is achieved in the order of seconds or minutes for the phase partitioning of relatively high volatility organic compounds into liquid particles, while the equilibration timescale ranges from hours to days for semi-solid viscous particles, low volatility species or large particle sizes, and thus can be subject to equilibrium shifts by faster chemical reactions in each phase. However, these considerations are beyond the scope of the study, as the goal was to determine how aqueous-phase chemistry can affect organic nitrates fate even at very low LWC. Thus, Eq. (12) was used to understand if the partition into the aerosol-phase is important compared to the aqueous-phase and the gas-phase, under the two investigated conditions."

**Can some of the organic nitrate be surface active and partition to the air-water interface?**

To our knowledge, there is no study on the role of organic nitrates as surfactants. Compounds with a significant surface activity present a polar group on one side and a saturated long carbon chain on the other. None of the organic nitrates presented in this work shows such a structure, except, perhaps, for 2-ethylhexyl nitrate. This may be the reason for the lack of solubility observed for this compound in our experiments.

**- Line 466, how did the authors consider the reactivities of carboxylic acid and carboxylate? Is this parameterized in the previous version of SAR?**

Yes. Parameters from Monod and Doussin (2008) were used to determine the $k_{OH,aq}$ of carboxylic acids and their conjugated bases.

To make clearer how $k_{OH,aq}$ values were calculated the manuscript was modified as follows:

In line 487: "The $k_{OH,aq}$ rate constants were calculated using the extended SAR developed in this work (Eq. (6)). Neighboring factors for nitrate and ether groups were those calculated in this work (Table 4) while all other parameters were taken from Doussin and Monod, (2013) and Monod and Doussin (2008). Eq. (8) was employed for compounds containing carbonyl groups. For organic nitrates containing a carboxylic acid functional group, the contributions of both the protonated molecule and its conjugated base were considered."

**Technical Comments**

**- Line 283, the sentence, "indicating a likely solvent kinetic effect which lowers more effectively the reaction." is awkward. Maybe consider rephrasing to "implying a solvent kinetic effect that can more effectively suppress the reactivity"?**

This has been modified as proposed, except for that we kept "lower" instead of "suppress".

**- In a few places "prior" should be "prior to" E.g., "Prior to each experiment." Check lines 114, 154, 161, 193**

This has been modified as proposed.

**REVIEWER 2**

»Major points«

**The manuscript text is overall fine, but would be strengthened if unnecessary portions were removed (e.g., the discussion of the "direct method" to determine rate constants on lines 60 - 63), repetition was eliminated (e.g., lines 291-292 repeat lines 286-288), and the text was edited for issues such as noun-verb agreement (e.g., "Measurement were performed. . ." on line 178). In addition, using parentheses to try to put the opposite case inside of a sentence makes for difficult reading, e.g., line 516: "For molecules which partition to the aqueous-phase, a positive (respectively negative) value of indicates that the aqueous-phase reactivity increases (respectively decreases) the atmospheric loss of the molecule." This should be revised.**

Modifications were done to correct mistakes, repetitions of sentences and to clarify confusing sentences.

**There are some issues with pH. First, the solution is listed as "pH < 3" (line 109), but the value should be closer to pH 2 since solutions contained 5 mM H2SO4 (line 764). This should be clarified. Second, I appreciate that an acidic solution is needed to keep Fe dissolved, but pH 2 is not relevant for most atmospheric waters, which are much less acidic. The low pH values used in the experiments are a bit concerning since past work has shown that the OH rate constant is sometimes pH dependent for classes of compounds that do not have a change in acid-base speciation across the pH range (e.g., Kroflic et al., PCCP, DOI: 10.1039/c9cp05533a).**

A pH < 3 is needed to produce ·OH radicals by the Fenton reaction (Neyens and Baeyens, 2003). The manuscript has been modified to specify that all experiments were performed at pH = 2.5 as directly measured (Micro pH electrode, Thermo).

Compounds with a pH-dependent relation discussed in (Kroflič et al., 2020) are phenol and catechol. For these molecules, the ·OH radical adds to a double bond of the aromatic ring and forms an intermediate adduct. In this type of reaction, Smith et al., (2015) (referring to Ashton et al., (1995), Heath and Valsaraj, (2015) and Land and Ebert, (1967)) have hypothesized that the intermediate adducts lifetimes might decrease by the effect of the acidity or the high ionic strength at pH 2 (relative to pH 5). Among our set of organic nitrates, none of them contains any double carbon bond, thus their ·OH-oxidation likely proceeds via H-abstractions, without any intermediate adduct. For species undergoing H abstraction, the Fenton reaction has already been employed to determine $k_{OH}$ values without significant discrepancies with studies performed at higher pH (e.g.: Buxton et al., (1988) and Herrmann et al., (2010)). This fact was also validated in this work by the determination of isopropanol and acetone $k_{OH}$ values.

This was added to the manuscript in line 110: "pH = 2.5 (Micro pH electrode, Thermo)"

**Third, the SAR lumps the current rate constants with 32 literature values without any tabulation or discussion of the pH values for the literature results. It's likely they were not done at pH 2, so it's not clear they can be considered along with the organic nitrates studied in this work. This should be examined and discussed.**

The SAR only includes compounds reacting with ·OH by H-abstraction at various pH values. For the reason mentioned above, we consider that their experimental values can be used together.

**The 2-ethylhexyl nitrate kinetic data looks suspicious (Figure 4) and produce a rate constant that is lower than expected and not included in the SAR (line 345). The authors (line 278) suggest that the experiments might have suffered from solubility issues, but they don't test this. As it currently stands, this compound is thus in publication purgatory: the data is reported in the manuscript,**

**but is not trusted. The authors need to take a stand on the compound, either (1) eliminate it entirely from the manuscript or (2) do additional experiments (e.g., at a lower concentration) to try to get good data.**

To perform additional experiments at lower concentrations where 2-ethylhexyl nitrate would be completely dissolved in water would mean to work at concentrations around $10^{-6} - 10^{-7}$ M. This would suppose further uncertainties due to experimental difficulties e.g.: concentrations close to the detection limits of the analyzers. Furthermore, as mentioned in our response to Reviewer 1, from the molecular structure of 2-ethylhexyl nitrate, we suspect this molecule to present some surface activity. In this case, 2-ethylhexyl nitrate would remain at the water-air interface, thus preventing from complete dilution at any concentration. A specific experimental protocol would thus have to be deployed, but this is beyond the scope of this study. For these reasons, we did not perform any additional experiment, and, as suggested, we have discarded its kinetic data and removed it from all figures and tables. However, we would rather not eliminate all mention of this compound in the manuscript to report the limitations likely due to surface activity. Furthermore, to our knowledge this compound has never been investigated before for PTR-MS measurements, thus we want to provide this new data.

In line 273: "Data for 2-ethylhexyl nitrate showed inconsistency compared to the other alkyl nitrates (significantly lower value in respect to its chemical structure). It is possible that even with a low initial concentration ($5 \cdot 10^{-5}$ mol L$^{-1}$), well below its solubility threshold ($1 \cdot 10^{-4}$ mol L$^{-1}$) the complete dissolution of the compound was not achieved when the reaction started, thus inducing an incorrect rate constant value. Furthermore, it is suspected, from its structure, to present some surface activity, thus preventing this compound to dissolve into the bulk water."

**The calculation and discussion of the lifetimes of the organic nitrates should be improved. (a) Fates other than OH (gas and aqueous) need to be better described. There is little discussion of hydrolysis, mostly just three lines about the rapid hydrolysis of compounds with the nitrate group on a tertiary carbon (lines 506-510). But don't other nitrates also have significant hydrolysis rate constants? This should be described and quantified.**

To our knowledge the hydrolysis of the nitrate group is extremely structure dependent, mostly occurring for tertiary or allylic organic nitrates (Darer et al., 2011; Jacobs et al., 2014). It is an acid-catalyzed nucleophilic substitution unimolecular reaction (SN1) where the nitrate group is firstly protonated and after its elimination, a molecule of water attacks the carbocation leading to the formation of an alcohol and nitric acid (Rindelaub et al., 2016). The formation of α- and β-pinene particulate organic nitrates has been extensively studied in simulation chambers, showing that the non-hydrolyzable fraction ranges from 68 to 91 % in mass (Takeuchi and Ng, 2019). As mentioned in our answer to Reviewer 1, in our study, control experiments were performed at pH = 2.5 for all organic nitrates within the time scale of the experiments. None of them showed any decay due to reaction with $H_2O_2$ or hydrolysis of the nitrate group, and we verified that none of them showed the formation of $NO_3^-$.

We added in the manuscript, in line 115: "Furthermore, control experiments at pH = 2.5 confirmed that none of the organic nitrates underwent hydrolysis at significant rates under our experimental conditions."

**In addition, isn't photolysis (in both phases) a significant sink for some of the nitrates with an aldehyde or ketone group? This should be discussed, especially in terms of how these other fates would alter the lifetimes that are presented.**

[revised manuscript text omitted]

**(b) The aqueous (and thus multiphase) lifetimes depend on the aqueous OH concentrations. The aqueous [OH] used in the manuscript (1E-14 M) is from a numerical model and is approximately 2 – 10 times higher than measurements. For example, Kaur et al. (2018) report aqueous [OH] of approximately 5E-15M in cloud drops and an extrapolated 1E-15 M in particle water. These model-measured differences, and their impact on calculated multiphase lifetimes, should be discussed.**

The ·OH radical concentration used $(1 \cdot 10^{-14} \, M)$ corresponds to the lowest modeled concentration value. Measured values reported by Kaur and Anastasio, (2017) correspond to ·OH concentrations in remote cloud droplets in winter. These concentrations are essentially extremely variable, and the value we used could match to those reported for cloud droplets in summer (Figure 1 in Arakaki et al., (2013)) or to those in cloud droplet in polluted areas. However, the possibility for lower $[\cdot OH]_{aq}$ has been discussed and included in the manuscript.

Herein are the changes done in the manuscript, in line 574: "This effect might be enhanced considering that aqueous ·OH concentrations might be lower than the one used in Eq. (14), i.e.: $1 \cdot 10^{-14}$ M. This value corresponds to aqueous concentrations of ·OH in a polluted/urban cloud droplet (Tilgner et al., 2013). Nevertheless, for remote cloud droplets or wet aerosols, the aqueous ·OH concentration might be 2 to 10 times lower (Arakaki et al., 2013). These lower concentrations would induce an increase of the ·OH-oxidation lifetimes of molecules partitioning into the aqueous-phase."

**(c) On first read, Figure 8 suggests that compounds with high W values (i.e., where most oxidation occurs in the aqueous phase) will have shorter lifetimes because of this aqueous sink. But this is not the case - for many of the compounds there is very little difference between gas-phase lifetime and multiphase (i.e., gas and PM) lifetime, as discussed only later with Figure 9. It would be helpful to incorporate the lifetimes into Figure 8, e.g., as symbol color or size, to more clearly show this. Showing the lifetimes in a figure would be very helpful, as this information is currently buried in Supplemental Table S5.**

As indicated in the manuscript, a W value lower than 1 implies that more than 50% of the target compound is consumed in the aqueous-phase. Therefore, a value higher than 1 means that the gas-phase ·OH-oxidation reaction is more significant than the aqueous-phase reaction. The W value does not give information on the multiphase lifetime, but a compound with a low W value (where most oxidation occurs in the aqueous-phase) will tend to have a longer lifetime when considering its reaction in both phases than in the gas-phase only. This is mostly due to the lower reactivity of the nitrate group towards ·OH in the aqueous-phase. The compounds presenting a small difference between gas-phase lifetime and multiphase lifetime are those that barely partition to the aqueous-phase.

Figure 8 incorporating the multiphase ·OH-oxidation lifetimes as a function in time would look as it follows (the marker size increases proportionally to the logarithm of the lifetime):

[Figure]

We have not decided to include the lifetimes in Figure 8 because it increases the complexity of the figure (already with 5 dimensions). Nevertheless, as exposed above, the multiphase lifetimes were more detailed for the different organic nitrate families focusing on the chemical properties which have an influence on the lifetime. We think that this is more appropriated than including the lifetimes as a marker size.

**»Other points«**

**Line 39. Hydrolysis to form nitrate is also an important sink for some nitrates, especially some isoprene-derived organic nitrates. Photolysis should be mentioned also.**

The sentence was rearranged to also specify processes where $NO_x$ are removed from the atmosphere. During their long-range transport, these compounds may: 1) release back $NO_x$ far from $NO_x$ sources via direct photolysis and/or ·OH-oxidation or 2) act as a definitive sink of atmospheric $NO_x$ by hydrolysis of the nitrate group and/or by deposition.

**l. 79. The absolute uncertainty on the k(OH+SCN-) rate constant might be larger than the target rate constant, but this doesn't matter: it is the relative – not absolute – uncertainty (i.e., 0.20/1.12) that is propagated in the relative rate calculation to determine the uncertainty on the target compound.**

We agree on this point. The sentence was removed.

**l. 125. To avoid any confusion that k(OH,X) is a pseudo-first order rate constant for loss of X, the authors should explicitly state that this is the second-order rate constant and give the units (M-1 s-1).**

This has been clarified.

**l. 135. The method works as long as the reference and target compounds are in Henry's law equilibrium in the reaction flask. For highly reactive compounds this condition might break**

**down. Based on an estimated characteristic time for aqueous mixing in the flask, is there an upper limit to the rate constants that can be determined under their [OH] conditions?**

There should not be any upper limit to determine high-rate constants because the developed method has the advantage to be very versatile: the reaction can be slowed down by both lowering the $Fe^{2+}$ or adding the solution at slower speed. The reaction is produced by adding a solution of $Fe^{2+}$ to the reactor using an automatic syringe which adds the solution continuously. By slowing the speed of the syringe, and using lower concentrations of $Fe^{2+}$, a highly reactive compound would be consumed in a slower way. By optimizing these parameters and with an appropriate reference compound, we expect that any molecule soluble enough can be studied using this method.

**l. 243. Where is the ". . .previous calculation of the quantity of OH radicals. . ." that is referenced here?**

This was an error.

We removed the sentence from the manuscript, and we have added the following one: "the ratios $k_{OH,X}/k_{OH,M}$ observed are significantly different from the gas-phase rate constant ratios, which are about 2 or 3 times higher."

**Section 3.2 Indicate how uncertainties were determined. Were they based on the standard deviation between replicates propagated with the uncertainty in the MeOH rate constant?**

Yes. This has been included in the manuscript: "Uncertainties were calculated by propagating the standard deviation of the replicates with the uncertainty of the methanol $k_{OH}$ rate constant. Then, to account for the small number of experiments performed for some molecules, the uncertainties are given by the confidence limits of 95 % given by the Student's t-distribution."

**l. 429 (and throughout). Typically, the liquid water content abbreviation "LWC" has no subscripted portion.**

$L_{WC}$ has been changed to LWC through the whole manuscript except for Equations 11 and 13 were $L_{WC}$ was kept for clarity in the equations.

**l. 461. The gas-phase units are shown here but not described until later. Since they're not standard, clarify here that [OH](gas) is in mol-OH / L-air and that the same volume unit is used for gas-phase rate constants.**

This has been clarified.

**l. 471. Both values of what?**

The simulated $k_{OH,gas}$ and $K_H$. It has been clarified in the manuscript.

**Table 2. Include the number of experiments for each compound in the table. Also indicate in the table that uncertainties are 95% CI.**

This was included.

**Figure 5 is a helpful comparison of data across compounds and phases.**

**Figure 8. Put the symbol for the isoprene-derived nitrates in the figure legend (along with the rest of the classes) rather than as a sentence in the caption.**

Since there are isoprene-derived nitrates in three different families (aldehydes, keto and with more than 2 different functions) we should incorporate three different symbols or create a new different one. We believe it is easier to specify that filled symbols correspond to an isoprene-derived nitrate and keeping the information related to their functionalization.

**Figure 9. Include another panel that shows the analogous results for the cloud LWC condition.**

Figure 9 already include cloud LWC content conditions. However, we have added another panel with wet aerosol LWC conditions.

A discussion over Figure 9b has been included in the manuscript in line 579: "Figure 9b shows that under wet aerosol conditions, only terpene nitrates ·OH-oxidation lifetimes are affected when considering the aqueous-phase. Under this LWC condition, the proposed terpene nitrates partition both in the aqueous- and the aerosol-phase (Figure 7b). In the aerosol-phase, not only the ·OH-oxidation kinetic rate constants may differ from the gas- and the aqueous-phase ones but also other oxidants can play a crucial role in the atmospheric loss of these compounds (Kaur and Anastasio, 2018; Manfrin et al., 2019). Further studies comprising this reactivity are thus needed to better understand the fate of organic nitrates in all atmospheric phases."

**The SI does not have a list of references.**

A bibliography section was added.

**Table S5 includes the multiphase lifetime for the particle LWC condition, but not for the cloud LWC condition. The latter should be included.**

Table S5 includes cloud LWC condition multiphase lifetime and gas-phase lifetime (no LWC). However, since Figure 9b shows that for most of the proposed organic nitrates their multiphase lifetime under wet aerosol conditions barely differs from the gas-phase lifetimes except for terpene nitrates. Therefore, we only have included the lifetime under particle LWC conditions for the latest compounds.